# PlayVirtual: Augmenting Cycle-Consistent Virtual Trajectories for Reinforcement Learning

**Tao Yu**[1]* **Cuiling Lan**[2]† **Wenjun Zeng**[2] **Mingxiao Feng**[1] **Zhizheng Zhang**[2] **Zhibo Chen**[1]†

[1]University of Science and Technology of China    [2]Microsoft Research Asia

yutao666@mail.ustc.edu.cn, {culan,wezeng}@microsoft.com

fmxustc@mail.ustc.edu.cn, zhizzhang@microsoft.com, chenzhibo@ustc.edu.cn

## Abstract

Learning good feature representations is important for deep reinforcement learning (RL). However, with limited experience, RL often suffers from data inefficiency for training. For un-experienced or less-experienced trajectories (*i.e.*, state-action sequences), the lack of data limits the use of them for better feature learning. In this work, we propose a novel method, dubbed PlayVirtual, which augments cycle-consistent virtual trajectories to enhance the data efficiency for RL feature representation learning. Specifically, PlayVirtual predicts future states in a latent space based on the current state and action by a dynamics model and then predicts the previous states by a backward dynamics model, which forms a trajectory cycle. Based on this, we augment the actions to generate a large amount of virtual state-action trajectories. Being free of groudtruth state supervision, we enforce a trajectory to meet the cycle consistency constraint, which can significantly enhance the data efficiency. We validate the effectiveness of our designs on the Atari and DeepMind Control Suite benchmarks. Our method achieves the state-of-the-art performance on both benchmarks. Our code is available at https://github.com/microsoft/Playvirtual.

## 1   Introduction

Deep reinforcement learning (RL) combines the powerful representation capacity of deep neural networks and the notable advantages of RL for solving sequential decision-making problems. It has made great progress in many complex control tasks such as video games [35, 46, 3], and robotic control [22, 54, 36]. Despite the success of deep RL, it faces the challenge of data/sample inefficiency when learning from high-dimensional observations such as image pixels from limited experience [27, 61, 29]. Fitting a high-capability feature encoder using only scarce reward signals is data inefficient and prone to suboptimal convergence [49]. Humans can learn to play Atari games in several minutes, while RL agents need millions of interactions [44]. However, collecting experience in the real world is often expensive and time-consuming. One may need several months to collect interaction data for robotic arms training [22] or be troubled by collecting sufficient patient data to train a healthcare agent [52]. Therefore, from another perspective, making efficient use of limited experience for improving data efficiency becomes vital for RL.

Many methods improve data efficiency by introducing auxiliary tasks with useful self-supervision to learn compact and informative feature representations, which better serves policy learning. Previous works have demonstrated that good auxiliary supervision can significantly improve agent learning, like leveraging image reconstruction [49], the prediction of future states [41, 13, 30, 40], maximizing Predictive Information [38, 1, 34, 42, 31], or promoting discrimination through contrastive learning

---

*This work was done when Tao Yu was an intern at Microsoft Research Asia.

†Corresponding Author.

35th Conference on Neural Information Processing Systems (NeurIPS 2021).

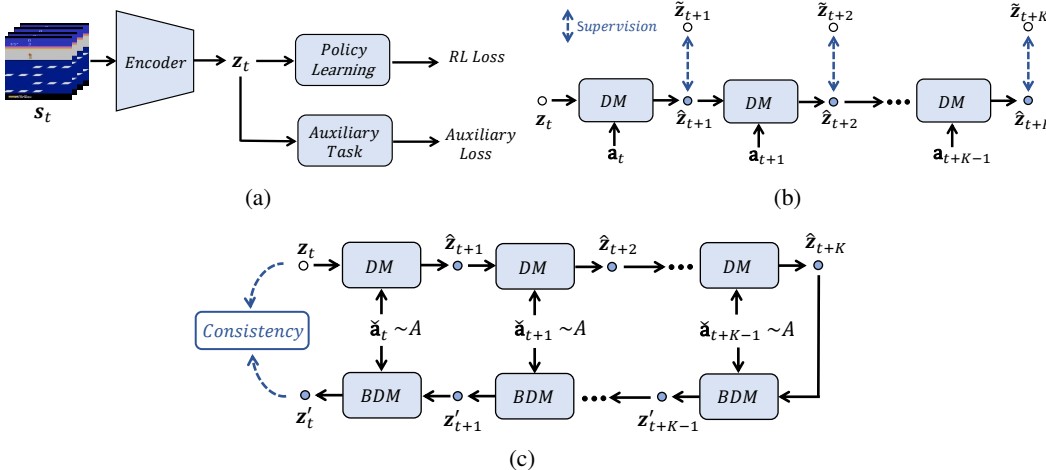

Figure 1: Illustration of the main pipeline of our method. (a) A glance at the overall framework which consists of an encoder for learning the latent state representation $\mathbf{z}_t$, a policy learning head, and our auxiliary task module. The auxiliary task module consumes a real trajectory as shown in (b) and an augmented virtual trajectory as shown in (c), respectively. In (b), we train the dynamics model (DM) to be predictive of the future state based on the input state and action, with the supervision from the future state. To enhance data efficiency, as shown in (c), we augment the actions to generate a virtual trajectory formed by a forward and a backward trajectory. Particularly, the forward state-action trajectory is obtained based on the current state $\mathbf{z}_t$, the DM and a sequence of augmented/generated actions (*e.g.*, for $K$ steps). Similarly, based on the predicted future state $\hat{\mathbf{z}}_{t+K}$, a backward dynamics model (BDM), and that sequence of augmented actions, we obtain the backward state-action trajectory. *For the virtual trajectory, we add the consistency constraint on the current state $\mathbf{z}_t$ and the predicted current state $\mathbf{z}'$ for optimizing the feature representations.*

[29, 59, 32, 25]. Although the above methods have been proposed to improve the data efficiency of RL, the limited experience still hinders the achievement of high performance. For instance, the current state-of-the-art method SPR [40] only achieves about $40\%$ of human level on Atari [2] when using data from 100k interactions with the environment. Some methods improve data efficiency by applying modest image augmentation (*i.e.*, transformations of the input images like random shifts and intensity) [28, 50]. Such perturbation on images improves the diversity of appearances of the input images. However, it cannot enrich the experienced trajectories (state-action sequences) in training and thus **the deep networks are still deficient in experiencing/ingesting flexible/diverse trajectories**.

In this work, to address the above problem, we propose a new method dubbed PlayVirtual, which augments cycle-consistent virtual trajectories to improve data efficiency. Particularly, we predict future states in the latent space by a dynamics model in the forward prediction (trained using real trajectories to predict the future state based on the current state and action) and then predict the previous states by a backward dynamics model, forming a loop. In this way, we can augment the actions to generate a large amount of virtual/fictitious state-action transitions for feature representation training, with the self-supervised cycle consistency constraint (which is a necessary/desirable condition for a good feature encoder). Note that such design is free of groundtruth supervision. Our augmentation to generate abundant virtual trajectories can significantly enhance the data efficiency. As illustrated in Figure 1, on top of a baseline RL framework, we introduce our PlayVirtual for augmenting state-action virtual trajectories with cycle consistency constraint. The dynamics model infers the future states recurrently based on the current state and a set of randomly sampled actions, and the backward dynamics model predicts the previous states according to the predicted future state and those sampled/augmented actions. We enforce the backwardly predicted state of the current time step to be similar to the original current state to meet the cycle consistency constraint.

We summarize the contributions of this work as follows:

- We pinpoint that augmenting the experience of RL in terms of trajectories is important for feature representation learning in RL (which is a sequential decision-making problem) to enhance data efficiency. To our best knowledge, we are the first to generate virtual trajectories (experience) for boosting feature learning.
- We propose a practical method PlayVirtual for augmenting virtual trajectories under self-supervised cycle consistency constraint, which significantly improves the data efficiency.

We demonstrate the effectiveness of our PlayVirtual on discrete control benchmark Atari [2], and continuous control benchmark DMControl Suite [43], where our PlayVirtual achieves the best performance on both benchmarks.

## 2   Related Work

### 2.1   Data/Sample-efficient RL and Representation Learning for RL

Learning from visual observations is a fundamental yet challenging problem in RL. In practice, collecting experience is expensive and time-consuming. Thus, in general only limited experience is available for training RL agent, which results in the difficulty in fitting a high-capability feature encoder, *i.e.*, learning powerful feature representations. Therefore, data-efficient RL has attracted a lot of attention and many methods are designed to make efficient use of limited experience for improving data efficiency. These methods can be grouped into three categories. (i) Auxiliary task based methods introduce auxiliary task to help representation learning of the states [21, 49, 41, 13, 30, 40, 38, 1, 34, 42, 31, 29, 59, 32]. (ii) Data augmentation based methods increase the diversity of image appearance through data augmentation [50, 28, 40]. But they do not augment virtual actions. (iii) World model based methods explicitly model the environment in order to enable the planning or promote the policy learning [16, 17, 61]. We only focus on the first two categories since the third one is not specifically designed to enhance the efficiency of feature representation learning.

In recent years, unsupervised representation learning has made significant progress in natural language processing [6, 33] and computer vision [38, 18, 5, 4, 12]. It aims to learn generic and discriminative feature representations without groudtruth supervision, *i.e.*, by introducing some unsupervised auxiliary tasks. In RL, a good state representation removes the redundancy and noise elements from the original high-dimensional state, and reshapes the original state space into a compact feature space. Recently, many works explore representation learning in RL and have shown promising performance improvement. UNREAL [21] introduces a number of auxiliary tasks such as reward prediction. Yarats *et al*. [49] introduce an auxiliary reconstruction loss to aid feature representation learning. Considering the ability to model what will happen next is necessary for success on many RL tasks [14, 31], some works train agents to be predictive of the future states. [41], PBL [13], SLAC [30] and SPR [40] explicitly predict the future states by modeling dynamics. Similarly, CPC [38], ST-DIM [1], DRIML [34], ATC [42] and PI-SAC [31] maximize the mutual information between the current state and the future state by using InfoNCE [38, 42], Deep InfoMax [20, 1, 34], or Conditional Entropy Bottleneck [9, 31]. Some works exploit contrastive learning to learn discriminative representations [29, 59, 32, 25]. CURL [29] extracts high-level features from raw pixels using contrastive learning, by encouraging the similarity between the augmentations of the same image and the dissimilarity between different images.

Inspired by the success of data augmentation in computer vision, DrQ [50] and RAD [28] explore the effectiveness of data augmentation in RL and show that increasing the diversity of the training images by simple image augmentation (such as random cropping) can improve the data-efficiency. In SPR [40], besides the prediction of its own latent state representations multiple steps into the future, they improve performance by adding data augmentation on the future input image as the future state supervision. However, all these methods train the encoder using the real interaction transitions. There is a lack of an efficient mechanism to generate reliable state-action pair transition trajectories for better training the feature encoder.

In this paper, we propose a method, dubbed PlayVirtual, which enables the augmentation of trajectories with unsupervised cycle consistency constraint for training the feature representation. This effectively enhances the data efficiency and our trajectory augmentation is conceptually orthogonal/complementary to the previous augmentation or auxiliary task based methods.

## 2.2 Cycle Consistency

Many works have explored the high-level idea of cycle consistency to address different challenges in various tasks such as image-to-image translation [60, 51, 24], image matching [56, 58, 57], feature representation learning [47, 7, 26]. For image-to-image translation, CycleGAN [60] introduces the cycle consistency constraint to Generative Adversarial Networks [10] to remove the requirement of groudtruth paired images for training, by enforcing the back-translated image to be the same as the original one. Zhang *et al.* [55] learn the correspondences that align dynamic robot behavior across two domains using cycle consistency: given observations in time $t$, the future prediction in time $t+1$ should be consistent across two domains under the consistent action taken. The purpose is to enable a direct transfer of the policy trained in one domain to the other without any additional fine-tuning. To learn visual correspondence from unlabeled video, Wang *et al.* [47] propose to track a target backward and then forward to meet a temporal cycle consistency on the feature representations, using the inconsistency between the start and end points as the loss function. Kong *et al.* [26] propose cycle-contrastive learning for learning video representations, which is designed to find correspondences across frames and videos considering the contrastive representation in their domains respectively, where the two domain representations form a video-frame-video cycle.

Different from the above works, in order to improve data efficiency in RL, we propose to augment virtual state-action trajectories to enrich the "experience" of the feature encoder for representation learning. To ensure the reasonbleness/correctness of the generated transitions/trajectories and make use of them, we take the necessary condition of a good trajectory, *i.e.*, satisfying the cycle consistency of the trajectory, as a constraint to optimize the network.

# 3 Cycle-Consistent Virtual Trajectories for Representation Learning in RL

## 3.1 Background

We consider reinforcement learning (RL) in the standard Markov Decision Process (MDP) setting where an agent interacts with the environment in episodes. We denote the state, the action taken by the agent and the reward received at timestep $t$ in an episode as $\mathbf{s}_t$, $\mathbf{a}_t$, and $r_t$, respectively. We aim to train an RL agent whose expected cumulative reward in each episode is maximized.

With the observation being high-dimensional short video clip at each timestep, the powerful representation capability of deep neural networks for encoding state and a strong RL algorithm contribute to the success of an RL agent. Similar to [29], we use the widely adopted RL algorithm Rainbow [19] for discrete control benchmarks (*e.g.*, Atari [2]) and Soft Actor Critic (SAC) [15] for continuous control benchmarks (*e.g.*, DMControl Suite [43]). Following SPR [40], we introduce a dynamics model (DM) to predict the future latent states multiple steps, which enables a forward state-action trajectory. We take SPR [40] as our baseline scheme.

## 3.2 Overall Framework

Considering the data efficiency in RL with limited experience, we propose a method named PlayVirtual to efficiently improve the feature representation learning of RL. Our key idea is to augment the actions to generate *virtual* state-action trajectories for boosting the representation learning of the encoder. Particularly, we eliminate the need of groudtruth trajectory supervision for the augmented sequences by using a cycle consistency constraint, which thus enhances data efficiency in training.

Figure 1 illustrates the main pipeline of our framework (with some details not presented for clarity). As shown in (a), it consists of an encoder which encodes the input observation $\mathbf{s}_t$ into low-dimensional latent state representation $\mathbf{z}_t$, an RL policy learning head (Rainbow [19] or SAC [15]), and our auxiliary task module. Particularly, as shown in (c), our auxiliary task module consists of a dynamics model (DM) which predicts future latent state based on the current state and the action, and a backward dynamics model (BDM) for backward state prediction. Following SPR [40], the DM is trained with the real state-action trajectory under the supervision of the future state (see (b)) to assure its capability of generating "correct" state transition. However, under limited experience, the encoder has few opportunities to be trained by those un-experienced or less-experienced state-action trajectories, which should be important to enhance data efficiency. To address this problem, as illustrated in (c), we add a BDM which predicts the previous state based on the current state and

the previous action. Together with the DM, the forward predictions and backward predictions form a cycle/loop, where the current state and the backwardly predicted current state are expected to be the same. Particularly, we augment the actions to generate virtual trajectories in order to train the network to "see" more flexible experiences with cycle consistency constraint. Our method contains three main components which we describe below.

**Dynamics Model for Prediction of Future States.** A one-step Markov transition $(\mathbf{s}_t, \mathbf{a}_t, \mathbf{s}_{t+1})$ contains a current state $\mathbf{s}_t \in \mathcal{S}$, an action $\mathbf{a}_t \in \mathcal{A}$, and the next state $\mathbf{s}_{t+1} \in \mathcal{S}$. The transition model determines the next state $\mathbf{s}_{t+1}$ given the current state-action pair $(\mathbf{s}_t, \mathbf{a}_t)$.

Considering the ability to model what will happen next is important for RL tasks, many works train agents to be predictive of the future states to learn good feature representations [40, 41, 13, 30]. Following SPR [40], we introduce a dynamics model (DM) $h(\cdot, \cdot)$ to predict the transition dynamics $(\mathbf{z}_t, \mathbf{a}_t) \rightarrow \mathbf{z}_{t+1}$ in the latent feature space, where $\mathbf{z}_t = f(\mathbf{s}_t)$ is encoded by the feature encoder $f(\cdot)$ of the current input video clip $\mathbf{s}_t$. As illustrated in Figure 1(b), based on the current input state $\mathbf{z}_t$ and a sequence of actions $\mathbf{a}_{t:t+K-1}$, we obtain a sequence of $K$ predictions $\hat{\mathbf{z}}_{t+1:t+K}$ of the future state representations using the action-conditioned transition model (*i.e.*, DM) $h(\cdot, \cdot)$ by computing the next state iteratively as

$$\hat{\mathbf{z}}_{t+k+1} = \begin{cases} h(\mathbf{z}_{t+k}, \mathbf{a}_{t+k}) & \text{if } k = 0 \\ h(\hat{\mathbf{z}}_{t+k}, \mathbf{a}_{t+k}) & \text{if } k = 1, 2, \cdots, K-1. \end{cases} \tag{1}$$

We train the DM with the supervision of the future state representations obtained from the recorded real trajectory (*i.e.*, from the recorded future video clip). Following SPR [40], we compute the prediction loss by summing over difference (error) between the predicted representations $\hat{\mathbf{z}}_{t+k}$ and observed representations $\tilde{\mathbf{z}}_{t+k}$ at timesteps $t + k$ for $1 \leq k \leq K$ measured in a "projection" metric space as:

$$\mathcal{L}_{pred} = \sum_{k=1}^{K} d(\hat{\mathbf{z}}_{t+k}, \tilde{\mathbf{z}}_{t+k}), \tag{2}$$

where $d$ denotes the distance metric in a "projection" space [40] (see Appendix A for more details).

This module has two roles in our framework. (i) The future prediction helps to learn good feature state representation, which enables the scheme SPR [40] that we use as our strong baseline. (ii) It paves the way for our introduction of cycle-consistent virtual trajectories for improving data efficiency.

**Backward Dynamics Model for Prediction of Previous States**: Backward transition model intends to determine the previous state $\mathbf{s}_t$ given the next state $\mathbf{s}_{t+1}$ and the causal action $\mathbf{a}_t$. We introduce a backward dynamics model (BDM) $b(\cdot, \cdot)$ to predict the backward transition dynamics $(\mathbf{z}_{t+1}, \mathbf{a}_t) \rightarrow \mathbf{z}_t$ in the latent feature space.

In previous works [11, 8, 37], backward induction has been exploited to predict the preceding states that terminate at a given high-reward state, where these traces of (state, action) pairs are used to improve policy learning. Their purpose is to emphasize the training on high-reward states and the probable trajectories leading to them to alleviate the problem of lack of high reward states for policy learning. In contrast, we introduce a BDM which predicts previous states (to have a backward trajectory) in order to build a cycle/loop with the forward trajectory to enforce the consistency constraint for boosting feature representation learning.

As illustrated in Figure 1(c), based on the hidden state $\hat{\mathbf{z}}_{t+K}$ and a sequence of actions $\check{\mathbf{a}}_{t+K-1:t}$, we obtain a sequence of $K$ predictions $\mathbf{z}'_{t+K-1:t}$ of the previous state representations using the BDM $b(\cdot, \cdot)$ by computing the previous state iteratively as

$$\mathbf{z}'_{t+k-1} = \begin{cases} b(\hat{\mathbf{z}}_{t+k}, \check{\mathbf{a}}_{t+k-1}) & \text{if } k = K \\ b(\mathbf{z}'_{t+k}, \check{\mathbf{a}}_{t+k-1}) & \text{if } k = K-1, K-2, \cdots, 1. \end{cases} \tag{3}$$

**Action Augmentation and Cycle Consistency Constraint.** Given the DM, BDM, a current state, and a sequence of actions, we can easily generate a forward trajectory and a corresponding backward trajectory which forms a loop/cycle/forward-backward trajectory. As we know, for an encoder which is capable of encoding observations to suitable feature representations, the feature representations of the start state $\mathbf{z}_t$ and the end state $\mathbf{z}'_t$ of a forward-backward trajectory should in general be similar/consistent, given a reasonable DM and BDM.

Therefore, as illustrated in Figure 1(c), we enforce a consistency constraint between the start state $\mathbf{z}_t$ and the end state $\mathbf{z}'_t$ to regularize the feature representation learning. In this way, by augmenting actions (generating/sampling virtual actions), we can obtain abundant virtual cycle-consistent trajectories for training. Note that in the training, we do not need supervision of the states from real trajectories.

Here, we mathematically define the cycle-consistent feature representation in a forward-backward trajectory as below.

**Definition 1.** *Given a (forward) dynamics model $h$ and a backward dynamics model $b$, cycle-consistent feature representation $\mathbf{z}_t$ in a forward-backward trajectory $\tau^c$ is a representation of the current state that meets the following condition when experiencing any sequence of $K$ actions $\{\mathbf{a}_t, \mathbf{a}_{t+1}, \ldots, \mathbf{a}_{t+K-1}\}$ sampled from an action space $\mathcal{A}$:*

$$\mathbb{E}_{\tau^c}[d_{\mathcal{M}}(\mathbf{z}'_t, \mathbf{z}_t)] = 0,$$

*where $d_{\mathcal{M}}$ is a distance metric on space $\mathcal{M}$ and $\mathbf{z}'_t$ is the prediction of $\mathbf{z}_t$ after experiencing a sequence of actions in forward prediction and backward prediction as*

$$forward: \ \hat{\mathbf{z}}_t = \mathbf{z}_t, \ \hat{\mathbf{z}}_{t+k+1} = h(\hat{\mathbf{z}}_{t+k}, \mathbf{a}_{t+k}), \ for \ k = 0, 1, \cdots, K-1,$$
$$backward: \ \mathbf{z}'_{t+K} = \hat{\mathbf{z}}_{t+K}, \ \mathbf{z}'_{t+k} = b(\mathbf{z}'_{t+k+1}, \mathbf{a}_{t+k}), \ for \ k = K-1, K-2, \cdots, 0.$$

Given the state $\mathbf{z}_t$ encoded from the current input $\mathbf{s}_t$ of time $t$, we randomly sample $M$ sets of actions in the action space $\mathcal{A}$. We calculate the cycle consistency loss as:

$$\mathcal{L}_{cyc} = \frac{1}{M} \sum_{m=1}^{M} d_{\mathcal{M}}(\mathbf{z}'_t, \mathbf{z}_t). \tag{4}$$

We describe the alternative distance metrics $d_{\mathcal{M}}$ on space $\mathcal{M}$ and study the influence on performance in Section 4.3.

*Discussion*: In our scheme, similar to [11, 8, 37], we model the backward dynamics using a BDM. This is basically feasible for many real-world applications, *e.g.*, robotic control, and many games. Consider a robotic arm: given a current state (*e.g.*, spatial position and rotation) and the previous action (*e.g.*, quaternions or six-degree-of-freedom parameters), one can deduce the previous state without much effort. This holds in most games such as chess or Atari. For some cases where there are many-to-one transition (*i.e.*, different states with the same action may be transited into the same state), BDM may learn the most probable state or a mean state to minimize the prediction errors through the powerful fitting capacity of neural networks. More discussion can be found in Appendix C.

**Overall Training Objective.** The overall training objective of our method is as below:

$$\mathcal{L}_{total} = \mathcal{L}_{rl} + \lambda_{pred}\mathcal{L}_{pred} + \lambda_{cyc}\mathcal{L}_{cyc}, \tag{5}$$

where $\mathcal{L}_{rl}$, $\mathcal{L}_{pred}$, and $\mathcal{L}_{cyc}$ denote the RL loss (please refer to Rainbow [19] for discrete control games, SAC [15] for continuous control games), prediction loss (see Eq. (2)), and cycle consistency loss (see Eq. (4)), respectively. $\lambda_{pred}$ and $\lambda_{cyc}$ are the hyperparameters for balancing the contributions of different losses.

## 4 Experiments

We introduce the experimental setup including environments, evaluation, and implementation details. We conduct extensive ablation studies to demonstrate and analyze the effectiveness of our designs.

### 4.1 Setup

**Environments and Evaluation.** We evaluate our method on the commonly used discrete control benchmark of Atari [2], and the continuous control benchmark of DMControl [43]. Following [29, 50], we measure the performance of different methods at 100k *interaction steps* (400k environment steps with action repeat of 4) on Atari (also refer to as **Atari-100k**), and at 100k and 500k *environment steps* on DMControl (refer to as **DMControl-100k** or **DMC-100k**, **DMControl-500k** or **DMC-500k**),

respectively. In general, using Atari-100k on 26 selected games [61, 45, 23, 29] and DMControl-100k [29, 50] has been a common practice for investigating data efficiency.

For Atari-100k, we measure the performance by score, and the median human-normalized score (*i.e.*, *median HNS*) of the 26 games. Human-normalized score on a game is calculated by $\frac{S_A - S_R}{S_H - S_R}$, where $S_A$ is the agent score, $S_R$ is the score of random play, and $S_H$ is the expert human score. For DMControl, the maximum possible score for every environment is $1000$ [43]. Following [29, 49, 16, 50, 32], we evaluate models on the six commonly used DMControl environments. Additionally, we use the *median score* on them to reflect the overall performance.

**Implementation Details.** For the discrete control benchmark of **Atari**, we use SPR [40] as our strong baseline (dubbed *Baseline*) and build our method on top of SPR by augmenting cycle-consistent virtual trajectories for better representation learning (see Figure 1(c)). For the backward dynamics model, we use the same architecture as that of the dynamics model. We set the number of prediction steps $K$ to 9 by default. We simply set the number of action sets, *i.e.*, the number of virtual trajectories $M$ to $2|\mathcal{A}|$, which is proportional to the size of action space $|\mathcal{A}|$ in that Atari game. To generate an action sequence, we randomly sample an action from the discrete action space at each step. *We report the results of SPR [40] by re-running their source code in all Tables except for Table 1 (results in which are copied from their paper, being higher than our reproduced results).*

For the continuous control benchmark of **DMControl**, considering the SPR is originally designed only for discrete control, we build a SPR-like scheme SPR$^\dagger$as our baseline (dubbed *Baseline*) for continuous control games. Particularly, we use the encoder and policy networks of CURL [29] as the basic networks. Following SPR [40], we remove the contrastive loss in CURL and introduce BYOL [12] heads to build SPR-like baseline scheme. We use the network architecture similar to the dynamics model in DBC [53] to build the dynamics model in SPR$^\dagger$ and the backward dynamics model in our PlayVirtual. We follow the training settings in CURL except the batch size (reduced from 512 to 128 to save memory cost) and learning rate. We set $K$ to 6, and set $M$ to a fixed number 10 with actions randomly sampled from the uniform distribution of the continuous action space.

We set $\lambda_{pred} = 1$ and $\lambda_{cyc} = 1$. For $d_{\mathcal{M}}$, we use the distance metric as in SPR [40]. More implementation details can be found in Appendix A. All our models are implemented via PyTorch [39].

### 4.2 Performance Comparison with State-of-the-Arts

Table 1: Scores achieved by different methods on Atari-100k. We also report median HNS. We run our PlayVirtual with 15 random seeds given that this benchmark is susceptible to high variance across multiple runs. Note that here we report the results of SPR [40] copied from their paper (*i.e.*, 41.5%), which is much higher than our reproduced results using their released source code (*i.e.*, 37.1%).

| Game | Human | Random | SimPLe[61] | DER[45] | OTR[23] | CURL[29] | DrQ[50] | SPR[40] | PlayVirtual |
|---|---|---|---|---|---|---|---|---|---|
| Alien | 7127.7 | 227.8 | 616.9 | 739.9 | 824.7 | 558.2 | 771.2 | 801.5 | **947.8** |
| Amidar | 1719.5 | 5.8 | 88.0 | **188.6** | 82.8 | 142.1 | 102.8 | 176.3 | 165.3 |
| Assault | 742.0 | 222.4 | 527.2 | 431.2 | 351.9 | 600.6 | 452.4 | 571.0 | **702.3** |
| Asterix | 8503.3 | 210.0 | **1128.3** | 470.8 | 628.5 | 734.5 | 603.5 | 977.8 | 933.3 |
| Bank Heist | 753.1 | 14.2 | 34.2 | 51.0 | 182.1 | 131.6 | 168.9 | **380.9** | 245.9 |
| Battle Zone | 37187.5 | 2360.0 | 5184.4 | 10124.6 | 4060.6 | 14870.0 | 12954.0 | **16651.0** | 13260.0 |
| Boxing | 12.1 | 0.1 | 9.1 | 0.2 | 2.5 | 1.2 | 6.0 | 35.8 | **38.3** |
| Breakout | 30.5 | 1.7 | 16.4 | 1.9 | 9.8 | 4.9 | 16.1 | 17.1 | **20.6** |
| Chopper Command | 7387.8 | 811.0 | **1246.9** | 861.8 | 1033.3 | 1058.5 | 780.3 | 974.8 | 922.4 |
| Crazy Climber | 35829.4 | 10780.5 | **62583.6** | 16185.3 | 21327.8 | 12146.5 | 20516.5 | 42923.6 | 23176.7 |
| Demon Attack | 1971.0 | 152.1 | 208.1 | 508.0 | 711.8 | 817.6 | 1113.4 | 545.2 | **1131.7** |
| Freeway | 29.6 | 0.0 | 20.3 | **27.9** | 25.0 | 26.7 | 9.8 | 24.4 | 16.1 |
| Frostbite | 4334.7 | 65.2 | 254.7 | 866.8 | 231.6 | 1181.3 | 331.1 | 1821.5 | **1984.7** |
| Gopher | 2412.5 | 257.6 | 771.0 | 349.5 | **778.0** | 669.3 | 636.3 | 715.2 | 684.3 |
| Hero | 30826.4 | 1027.0 | 2656.6 | 6857.0 | 6458.8 | 6279.3 | 3736.3 | 7019.2 | **8597.5** |
| Jamesbond | 302.8 | 29.0 | 125.3 | 301.6 | 112.3 | **471.0** | 236.0 | 365.4 | 394.7 |
| Kangaroo | 3035.0 | 52.0 | 323.1 | 779.3 | 605.4 | 872.5 | 940.6 | **3276.4** | 2384.7 |
| Krull | 2665.5 | 1598.0 | **4539.9** | 2851.5 | 3277.9 | 4229.6 | 4018.1 | 3688.9 | 3880.7 |
| Kung Fu Master | 22736.3 | 258.5 | **17257.2** | 14346.1 | 5722.2 | 14307.8 | 9111.0 | 13192.7 | 14259.0 |
| Ms Pacman | 6951.6 | 307.3 | **1480.0** | 1204.1 | 941.9 | 1465.5 | 960.5 | 1313.2 | 1335.4 |
| Pong | 14.6 | -20.7 | **12.8** | -19.3 | 1.3 | -16.5 | -8.5 | -5.9 | -3.0 |
| Private Eye | 69571.3 | 24.9 | 58.3 | 97.8 | 100.0 | **218.4** | -13.6 | 124.0 | 93.9 |
| Qbert | 13455.0 | 163.9 | 1288.8 | 1152.9 | 509.3 | 1042.4 | 854.4 | 669.1 | **3620.1** |
| Road Runner | 7845.0 | 11.5 | 5640.6 | 9600.0 | 2696.7 | 5661.0 | 8895.1 | **14220.5** | 13534.0 |
| Seaquest | 42054.7 | 68.4 | **683.3** | 354.1 | 286.9 | 384.5 | 301.2 | 583.1 | 527.7 |
| Up N Down | 11693.2 | 533.4 | 3350.3 | 2877.4 | 2847.6 | 2955.2 | 3180.8 | **28138.5** | 10225.2 |
| Median HNS (%) | 100 | 0 | 14.4 | 16.1 | 20.4 | 17.5 | 26.8 | 41.5 | **47.2** |

Table 2: Scores (mean and standard deviation) achieved by different methods on the DMControl-100k and DMControl-500k. We run our PlayVirtual with 10 random seeds. Note that SPR [40] is originally designed only for discrete control. For the continuous-control environments, we extend SPR to a new version named SPR$^{\dagger}$.

| 100k Step Scores | PlaNet[16] | Dreamer[17] | SAC+AE[49] | SLAC[30] | CURL[29] | DrQ [50] | SPR$^{\dagger}$[40] | PlayVirtual |
|---|---|---|---|---|---|---|---|---|
| Finger, spin | $136 \pm 216$ | $341 \pm 70$ | $740 \pm 64$ | $693 \pm 141$ | $767 \pm 56$ | $901 \pm 104$ | $868 \pm 143$ | $\mathbf{915 \pm 49}$ |
| Cartpole, swingup | $297 \pm 39$ | $326 \pm 27$ | $311 \pm 11$ | - | $582 \pm 146$ | $759 \pm 92$ | $799 \pm 42$ | $\mathbf{816 \pm 36}$ |
| Reacher, easy | $20 \pm 50$ | $314 \pm 155$ | $274 \pm 14$ | - | $538 \pm 233$ | $601 \pm 213$ | $638 \pm 269$ | $\mathbf{785 \pm 142}$ |
| Cheetah, run | $138 \pm 88$ | $235 \pm 137$ | $267 \pm 24$ | $319 \pm 56$ | $299 \pm 48$ | $344 \pm 67$ | $467 \pm 36$ | $\mathbf{474 \pm 50}$ |
| Walker, walk | $224 \pm 48$ | $277 \pm 12$ | $394 \pm 22$ | $361 \pm 73$ | $403 \pm 24$ | $\mathbf{612 \pm 164}$ | $398 \pm 165$ | $460 \pm 173$ |
| Ball in cup, catch | $0 \pm 0$ | $246 \pm 174$ | $391 \pm 82$ | $512 \pm 110$ | $769 \pm 43$ | $913 \pm 53$ | $861 \pm 233$ | $\mathbf{926 \pm 31}$ |
| Median Score | 137.0 | 295.5 | 351.0 | 436.5 | 560.0 | 685.5 | 719.0 | **800.5** |
| **500k Step Scores** | | | | | | | | |
| Finger, spin | $561 \pm 284$ | $796 \pm 183$ | $884 \pm 128$ | $673 \pm 92$ | $926 \pm 45$ | $938 \pm 103$ | $924 \pm 132$ | $\mathbf{963 \pm 40}$ |
| Cartpole, swingup | $475 \pm 71$ | $762 \pm 27$ | $735 \pm 63$ | - | $841 \pm 45$ | $868 \pm 10$ | $\mathbf{870 \pm 12}$ | $865 \pm 11$ |
| Reacher, easy | $210 \pm 390$ | $793 \pm 164$ | $627 \pm 58$ | - | $929 \pm 44$ | $942 \pm 71$ | $925 \pm 79$ | $\mathbf{942 \pm 66}$ |
| Cheetah, run | $305 \pm 131$ | $570 \pm 253$ | $550 \pm 34$ | $640 \pm 19$ | $518 \pm 28$ | $660 \pm 96$ | $716 \pm 47$ | $\mathbf{719 \pm 51}$ |
| Walker, walk | $351 \pm 58$ | $897 \pm 49$ | $847 \pm 48$ | $842 \pm 51$ | $902 \pm 43$ | $921 \pm 45$ | $916 \pm 75$ | $\mathbf{928 \pm 30}$ |
| Ball in cup, catch | $460 \pm 380$ | $879 \pm 87$ | $794 \pm 58$ | $852 \pm 71$ | $959 \pm 27$ | $963 \pm 9$ | $963 \pm 8$ | $\mathbf{967 \pm 5}$ |
| Median Score | 405.5 | 794.5 | 764.5 | 757.5 | 914.0 | 929.5 | 920.0 | **935.0** |

**Comparison on Atari.** On Atari-100k, Table 1 shows the comparisons with the state-of-the-art methods. We also report the results of random play (Random) and expert human play (Human) (copied from [48]). PlayVirtual achieves a median HNS of 47.2%, significantly outperforming all previous methods. PlayVirtual surpasses the baseline SPR [40](with a median HNS of 41.5% reported in their paper) by **5.7%**. We have re-run the released source code of SPR with 15 random seeds and obtain a median HNS of 37.1%, which suggests that our improvement over SPR is actually **10.1%**.

**Comparison on DMControl.** For each environment in DMControl, we run our PlayVirtual with 10 random seeds to report the results. Table 2 shows the comparisons with the state-of-the-art methods. Our method performs the best for the majority (**5** out of **6**) of the environments on both DMControl-100k and DMControl-500k. (i) On DMControl-100k which is in low data regime, our method achieves the highest median score of 800.5, which is about **11.3%** higher than SPR$^{\dagger}$, 16.7% higher than DrQ [50] and 42.9% higher than CURL [29]. (ii) On DMControl-500k, our method achieves a median score of 935.0, which approaches the perfect score of 1000 and outperforms all other methods. Therefore, our method achieves superior performance in both data-efficiency and asymptotic performance.

### 4.3 Ablation Studies

We use the median HNS and median score to measure the overall performance on Atari and DMControl, respectively. We run each game in Atari with 15 random seeds. To save computational resource, we run each environment in DMControl with 5 random seeds (instead of 10 as in Table 2).

**Effectiveness of PlayVirtual.** As described in Section 4.1, we take SPR [40] as our baseline (*i.e.*, *Baseline*) on discrete control benchmark Atari, and SPR$^{\dagger}$ on continuous control benchmark DMControl. On top of *Baseline*, we validate the effectiveness of our PlayVirtual which augments cycle-consistent virtual trajectories for improving data efficiency. Table 3 shows the comparisons. We can see that *PlayVirtual* achieves a median HNS of 47.2%, which outperforms *Baseline* by **10.1%** on Atari-100k. On DMControl-100, *PlayVirtual* improves *Baseline* from 728.0 to 797.0 in terms of median score (*i.e.*, a relative gain of 9.5%). As a comparison, *Baseline* outperforms *Baseline w/o Pred* by 3.7% on Atari-100k, where "Pred" denotes the prediction of future state in SPR / SPR$^{\dagger}$(*i.e.*, the contribution of SPR [40]). The large gains of our PlayVirtual over *Baseline* demonstrate the effectiveness of our PlayVirtual in boosting feature

Table 3: Effectiveness of PlayVirtual on top of *Baseline*, which is SPR [40] for discrete control on Atari, and SPR$^{\dagger}$for continual control on DMControl. "w/o Pred" denotes disabling future prediction in *Baseline*. *Baseline+BDM* denotes the scheme that a BDM is incorporated into *Baseline*.

| Model | Atari-100k | DMControl-100k |
|---|---|---|
| Baseline w/o Pred | 33.4 | 680.0 |
| Baseline | 37.1 | 728.0 |
| Baseline+BDM | 38.4 | 741.0 |
| PlayVirtual | **47.2** | **797.0** |

representation learning. In addition, to further benchmark PlayVirtual's data efficiency, we compare the testing performance in every 5k environment steps at the first 100k on DMControl, where the result curves in Appendix B show that our PlayVirtual consistently outperforms *Baseline*.

One may wonder whether the major performance gain of our PlayVirtual is attributed to the introduction of backward dynamics model (BDM) or by our augmentation of virtual trajectories. When we disable the augmentation of virtual trajectories, our scheme degrades to *Baseline+BDM*, where a BDM is incorporated into the baseline SPR (or SPR$^\dagger$) and only the real trajectories go through the BDM. In Table 3, we can see that introducing BDM does not improve the performance obviously and our augmentation of cycle-consistent virtual trajectories for regularizing feature representation learning is the key for the success.

**Influence of Prediction Steps $K$.** We study the influence of $K$ for both our PlayVirtual and the baseline scheme SPR/SPR$^\dagger$. Table 4 shows the performance. When $K = 0$, both schemes degrade to *Baseline w/o Pred* (where future prediction is disabled in SPR/SPR$^\dagger$). We have the following observations/conclusions. (i) Given the same number of prediction steps $K$ (beside

Table 4: Influence of prediction steps $K$ for our PlayVirtual and the baseline scheme SPR/SPR$^\dagger$.

| Benchmark | Model | $K$=0 | $K$=3 | $K$=6 | $K$=9 | $K$=12 |
|---|---|---|---|---|---|---|
| Atari-100k | SPR | 33.4 | 33.9 | 35.2 | 37.1 | 34.9 |
| | PlayVirtual | 33.4 | 34.8 | 39.2 | **47.2** | 43.1 |
| DMC-100k | SPR$^\dagger$ | 664.0 | 725.0 | 723.0 | 728.0 | 721.5 |
| | PlayVirtual | 664.0 | 775.5 | **797.0** | 795.0 | 794.5 |

0), our PlayVirtual consistently outperforms the baseline scheme SPR/SPR$^\dagger$ on both benchmarks Atari-100k/MDControl-100k. (ii) Our PlayVirtual achieves the best performance at $K = 9$ on Atari and $K = 6$ on DMControl, which outperforms the baseline at the same $K$ by 10.1% and 9.5% (relative gain) on Atari and DMControl, respectively. Note that the performance of SPR [40] obtained using their source code at $K$=5 (note $K$=5 is used in SPR paper) is 36.1% (which is 41.5% in their paper) on Atari-100k. (iii) In SPR [40]/SPR$^\dagger$, a too small number of prediction steps cannot make the feature representation sufficiently predictive of the future while a too large number of prediction steps may make the RL loss contributes less to the feature representation learning (where a more elaborately designed weight $\lambda_{pred}$ is needed). Our PlayVirtual follows similar trends.

**What does Augmenting Cycle-Consistent Virtual Trajectories Help?** We propose the augmentation of cycle-consistent virtual trajectories in order to boost the feature representation learning of RL for improving data efficiency. In the training, the cycle consistency loss $L_{cyc}$ over the virtual trajectories would optimize the parameters of the encoder, DM and BDM. One may wonder what the gain is mainly coming from. Is it because the DM is more powerful/accurate that enables better prediction of future states?

Table 5: Impact of the augmentation of cycle-consistent virtual trajectories on feature representation learning. *PlayVirtual-ND* denotes that we do not use the cycle consistency loss over virtual trajectories to update the dynamic model.

| Model | Atari-100k | DMControl-100k |
|---|---|---|
| Baseline | 37.1 | 723.0 |
| PlayVirtual-ND | 44.0 | 777.5 |
| PlayVirtual | **47.2** | **797.0** |

Or is it because the encoder becomes more powerful to provide better feature representation? We validate this by letting the cycle consistency loss $L_{cyc}$ not update DM, where DM is only optimized by prediction loss $L_{pred}$ as in SPR. We denote this scheme as *PlayVirtual-ND*. Table 5 shows that we obtain a gain of 6.9% in *PlayVirtual-ND* from the regularization of $L_{cyc}$ *on the encoder* and a gain of 10.1% in *PlayVirtual* from the regularization *on both the encoder and DM* on Atari. Similar trend is observed on DMControl. This implies that the augmentation of cycle-consistent virtual trajectories is helpful to DM training but the main gain is brought by its regularization on the feature representation learning of the encoder.

**Influence of Distance Metric $d_{\mathcal{M}}$ on Space $\mathcal{M}$.** For the distance metric $d_{\mathcal{M}}$ in space $\mathcal{M}$, we compare cosine distance on the latent feature space $\mathcal{M}_{latent}$, *i.e.*, $d_{\mathcal{M}}(\mathbf{z}'_t, \tilde{\mathbf{z}}_t) = 2 - 2\frac{\mathbf{z}'_t}{\|\mathbf{z}'_t\|}\frac{\tilde{\mathbf{z}}_t}{\|\tilde{\mathbf{z}}_t\|}$ and on the "projection" space $\mathcal{M}_{proj}$ as in SPR [40] (see Appendix A for more details). We compare the influence of feature space for calculating cycle consistency loss and show the results in Table 6. On the Atari benchmark, our PlayVirtual with distance metric on space $\mathcal{M}_{latent}$ and with distance metric on space $\mathcal{M}_{proj}$ significantly outperforms *Baseline* by 7.7% and 10.1%, respectively. This demonstrates the effectiveness of our key idea of exploiting virtual trajectories for effective representation learning. $\mathcal{M}_{proj}$ performs 2.4% better than $\mathcal{M}_{latent}$. That maybe because for PlayVirtual and *Baseline* for Atari, latent feature $\mathbf{z}_t$ (which corresponds to a $64 \times 7 \times 7$ feature map) preserves more spatial information than projected feature, where the former is less robust to

be matched across two augmented observations due to spatial misalignment. On the DMControl benchmark, our PlayVirtual with distance metric on space $\mathcal{M}_{latent}$ and with distance metric on space $\mathcal{M}_{proj}$ significantly outperforms *Baseline* by 70.5 and 69.0 in terms of median score, respectively. The performance of $\mathcal{M}_{latent}$ and $\mathcal{M}_{proj}$ are similar. Note that the latent feature $\mathbf{z}_t$ of PlayVirtual or *Baseline* (built based on CURL) corresponds to a feature vector which is obtained after a fully connected layer in the backbone network, which does not face the spatial misalignment problem caused by the augmentation. We use $\mathcal{M}_{proj}$ as the default metric space in this work.

Table 6: Influence of distance metric space $\mathcal{M}$. $\mathcal{M}_{latent}$ and $\mathcal{M}_{proj}$ denote the use of the latent feature space and the "projection" space, respectively.

| Model | Atari-100k | DMControl-100k |
|---|---|---|
| Baseline | 37.1 | 728.0 |
| PlayVirtual($\mathcal{M}_{latent}$) | 44.8 | **798.5** |
| PlayVirtual($\mathcal{M}_{proj}$) | **47.2** | 797.0 |

**Influence of the Number of Virtual Trajectories $M$.** Table 7 shows the influence of the number of virtual trajectories $M$. We can observe that small $M$ (less generated virtual trajectories) is inferior to a suitable $M$. That may be because too small $M$ cannot cover diverse experiences for feature representation learning. When $M$ is too large, it brings less additional benefit. That may be because a suitable number of trajectories is enough for regularizing the network training. We observe that the performance drops when $M$ is too large. That may be because a very large $M$ would increase the optimization difficulty in practice.

Table 7: Influence of the number of virtual trajectories $M$.

| Atari-100k | | | | | |
|---|---|---|---|---|---|
| M | 0 | $|\mathcal{A}|$ | $2|\mathcal{A}|$ | $3|\mathcal{A}|$ | |
| Median HNS(%) | 37.1 | 39.5 | **47.2** | 42.5 | |
| **DMControl-100k** | | | | | |
| M | 0 | 1 | 10 | 20 | 30 |
| Median Score | 723.0 | 770.5 | 797.0 | **806.0** | 792 |

## 5 Conclusion

With limited experience, deep RL suffers from data inefficiency. In this work, we propose a new method, dubbed PlayVirtual, which augments cycle-consistent virtual state-action trajectories to enhance the data efficiency for RL feature representation learning. PlayVirtual predicts future states based on the current state and a sequence of sampled actions and then predicts the previous states, which forms a trajectory cycle/loop. We enforce the trajectory to meet the cycle consistency constraint to regularize the feature representation learning. Experimental results on both the discrete control benchmark Atari and continuous control benchmark DMControl demonstrate the effectiveness of our method, where we achieve the state-of-the-art performance on both benchmarks.

## Acknowledgments and Disclosure of Funding

This work was supported in part by the National Key Research and Development Program of China 2018AAA0101400 and NSFC under Grant U1908209, 61632001 and 62021001.

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
