# PlayVirtual: Augmenting Cycle-Consistent Virtual Trajectories for Reinforcement Learning (Appendix)

**Tao Yu**[1][*] **Cuiling Lan**[2][†] **Wenjun Zeng**[2] **Mingxiao Feng**[1] **Zhizheng Zhang**[2] **Zhibo Chen**[1][†]
[1]University of Science and Technology of China    [2]Microsoft Research Asia
yutao666@mail.ustc.edu.cn, {culan,wezeng}@microsoft.com
fmxustc@mail.ustc.edu.cn, zhizzhang@microsoft.com, chenzhibo@ustc.edu.cn

## A    More Implementation Details

### A.1    Network Architecture

**Network Architecture for Discrete Control Benchmark of Atari.** For the discrete control benchmark of **Atari**, we use SPR [9] as our strong baseline (dubbed *Baseline*) and build our method on top of SPR by augmenting cycle-consistent virtual trajectories for better representation learning.

SPR [9] has three main components: (online) encoder $f(\cdot)$, dynamics model (DM) $h(\cdot, \cdot)$, and policy learning (Q-learning) head $\pi(\cdot)$. The encoder consists of three convolutional layers with ReLU layer after each convolutional layer. The DM is composed of two convolutional layers with batch normalization [5] after the first convolutional layer and ReLU after the second convolutional layer. The Q-learning head is designed following Rainbow [4]. Rather than predicting representations produced by the online encoder (by the DM), SPR computes target representations for future states using a target encoder $f_m$, whose parameters are an exponential moving average (EMA) of the online encoder parameters. To obtain the "projection" metric space $d$ (see Eq. (ii) in the main manuscript) for future state prediction optimization, SPR uses online and target projection heads $g(\cdot)$ and $g_m(\cdot)$ to project online and target representations to a smaller latent space, and apply a prediction head $q(\cdot)$ to the online projections to predict the target projections.

For our PlayVirtual, on top of SPR, we add a backward dynamics model (BDM) $b(\cdot, \cdot)$. For simplicity, we use the same network architecture as the DM. To calculate the cycle consistency loss for the feature representations (in a forward-backward trajectory) in a distance metric on space $\mathcal{M}$, we can simply use the cosine distance on the latent feature space, *i.e.*, $d_{\mathcal{M}}(\mathbf{z}'_t, \mathbf{z}_t) = 2 - 2 \frac{\mathbf{z}'_t}{\|\mathbf{z}'_t\|} \frac{\mathbf{z}_t}{\|\mathbf{z}_t\|}$. As a design alternative, we can use the "projection" metric space as in SPR [9] (discussed in the last paragraph) to calculate the cosine distance on the projection space, *i.e.*, $d_{\mathcal{M}}(\mathbf{z}'_t, \mathbf{z}_t) = 2 - 2 \frac{q(g(\mathbf{z}'_t))}{\|q(g(\mathbf{z}'_t))\|} \frac{g_m(\mathbf{z}_t)}{\|g_m(\mathbf{z}_t)\|}$. In our implementation, we could directly use $\mathbf{z}_t$ (the start state of the virtual trajectory) as the target feature representation. Motivated by SPR, for each trajectory, we use the feature representation $\tilde{\mathbf{z}}_t$ of a stochastic augmentation $\tilde{\mathbf{s}}_t$ of the current video clip (observation) $\mathbf{s}_t$, as the target feature representation. Then, $d_{\mathcal{M}}(\mathbf{z}'_t, \tilde{\mathbf{z}}_t)$ is the actual distance metric.

**Network Architecture for Continuous Control Benchmark of DMControl.** For the continuous control benchmark of **DMControl**, considering the SPR is originally designed only for discrete control, we build a SPR-like scheme SPR[†]as our baseline (dubbed *Baseline*) for continuous control games. Particularly, we use the encoder and policy networks of CURL [7] as the basic networks. Following SPR [9], we remove the contrastive loss in CURL and introduce BYOL [3] heads to build SPR-like baseline scheme. We use the network architecture similar to the dynamics model in DBC [11] to build the dynamics model (DM) in SPR[†], where the DM consists of two fully connected layers with an LN (layer normalization) layer and a ReLU after the first fully connected layer. The

---

[*]This work was done when Tao Yu was an intern at Microsoft Research Asia.
[†]Corresponding Author.

35th Conference on Neural Information Processing Systems (NeurIPS 2021).

---

**Algorithm 1** Training Algorithm for PlayVirtual

---

**Require:** denote parameters of an encoder $f$, a dynamics model $h$, a backward dynamics model $b$ and a policy learning head $\pi$, as $\theta_f, \xi_h, \xi_b$ and $\omega$, respectively;

1: denote the number of prediction steps as $K$, the number of virtual trajectories as $M$;
2: denote the prediction loss weight and the predefined maximum weight for cycle consistency loss as $\lambda_{pred}$ and $\lambda_{cyc}^{max}$, respectively;
3: denote the warmup end iteration as $i_{end}$;.
4: denote the replay buffer as $\mathcal{D}$;
5: denote the interaction step index for Atari and the environment step index for DMControl as $i$;
6: randomly initialize all network parameters and make the reply buffer empty.
7: **while** $train$ **do**
8:      determine the action $\mathbf{a} \sim \pi(f(\mathbf{s}))$ (based on policy) and interact with environment
9:      record/collect experience $\mathcal{D} \leftarrow \mathcal{D} \cup (\mathbf{s}, \mathbf{a}, \mathbf{s}_{next}, r)$
10:      sample a sequence of $(\mathbf{s}, \mathbf{a}, \mathbf{s}_{next}, r) \sim \mathcal{D}$
11:      $\mathcal{L}_{cyc} \leftarrow 0; \mathcal{L}_{pred} \leftarrow 0; \mathcal{L}_{rl} \leftarrow 0$
12:      $\mathbf{z}_t \leftarrow f(\mathbf{s}_t)$
13:      **for** $j = 1, 2, ..., M$ **do**
14:          $\{\check{\mathbf{a}}_t^{(j)}, \check{\mathbf{a}}_{t+1}^{(j)}, \ldots, \check{\mathbf{a}}_{t+K-1}^{(j)}\} \sim \mathcal{A}$            ▷ randomly sample a sequence of actions
15:          $\hat{\mathbf{z}}_t^{(j)} \leftarrow \mathbf{z}_t$
16:          **for** $k = 0, 1, ..., K-1$ **do**
17:              $\hat{\mathbf{z}}_{t+k+1}^{(j)} \leftarrow h(\hat{\mathbf{z}}_{t+k}^{(j)}, \check{\mathbf{a}}_{t+k}^{(j)})$            ▷ (forward) dynamics prediction
18:          **end for**
19:          $\mathbf{z}_{t+K}'^{(j)} \leftarrow \hat{\mathbf{z}}_{t+K}^{(j)}$
20:          **for** $k = K-1, K-2, ..., 0$ **do**
21:              $\mathbf{z}_{t+k}'^{(j)} \leftarrow b(\mathbf{z}_{t+k+1}'^{(j)}, \check{\mathbf{a}}_{t+k}^{(j)})$           ▷ backward dynamics prediction
22:          **end for**
23:          $\mathcal{L}_{cyc} \leftarrow \mathcal{L}_{cyc} + d(\mathbf{z}_t'^{(j)}, \mathbf{z}_t^{(j)})$            ▷ calculate cycle-consistency loss
24:      **end for**
25:      $\mathcal{L}_{cyc} \leftarrow \mathcal{L}_{cyc}/M$
26:      calculate the forward prediction loss $\mathcal{L}_{pred}$ according to Eq. (2)
27:      calculate the RL loss $\mathcal{L}_{rl}$
28:      warmup $\lambda_{cyc}$ based on $\lambda_{cyc}^{max}, i_{end}, i$
29:      $\mathcal{L}_{total} \leftarrow \mathcal{L}_{rl} + \lambda_{pred}\mathcal{L}_{pred} + \lambda_{cyc}\mathcal{L}_{cyc}$
30:      $\theta_f, \xi_h, \xi_b, \omega \leftarrow Optimize((\theta_f, \xi_h, \xi_b, \omega), \mathcal{L}_{total})$
31: **end while**

---

encoder has four convolutional layers (with a ReLU after each), followed by a fully connected layer, an LN layer [1], and a hyperbolic tangent (tanh) activation. Similar to the design in SPR, we have a projection head $g(\cdot)$, a prediction head $q(\cdot)$ for the (online) encoder, and a momentum encoder $f_m(\cdot)$ and a momentum projection head $g_m(\cdot)$. The projection head and prediction head are both built by two fully connected layers (with a ReLU layer after the first) of 512 hidden units for each.

For our PlayVirtual, we add a backward dynamics model (BDM) $b(\cdot, \cdot)$ which has the same architecture as the DM. We have the same design as in the discrete control case of the distance metric $d_{\mathcal{M}}$ on space $\mathcal{M}$.

### A.2 Training Details

**Training Algorithm.** We describe the main training procedure in Algorithm 1. Note that for the convenience of description, we parameterize the encoder $f$, dynamics model $h$, backward dynamics model $b$, and policy $\pi$ with $\theta_f, \xi_h, \xi_b$, and $\omega$, respectively.

**Hyperparameters.** We present the hyperparameters used for benchmarks of Atari and DMControl in Table 8 and 9, respectively. We set them mainly following SPR [9] on Atari, and CURL [7] on DMControl.

**Loss Details.** Our total loss is composed of three components: RL loss $\mathcal{L}_{rl}$, prediction loss $\mathcal{L}_{pred}$ and cycle loss $\mathcal{L}_{cyc}$. The RL loss is only applied on real trajectories to update the encoder and the

policy learning head. The prediction loss is applied on real trajectories to update the encoder and the DM. The cycle consistency loss acts only on virtual trajectories to update the encoder, the DM and the BDM. Note that we experimentally observe that additionally applying the cycle consistency loss on the real trajectories achieves only slight further improvement. For example, it achieves 0.1% improvement on Atari in the median human-normalized score (*i.e.*, median HNS).

**Warmup Scheme.** In the early stage of training, the dynamics model has not been trained well and thus the cycle-consistency constraint may not be reliable. Therefore, inspired by [6, 8], we ramp up the weight $\lambda_{cyc}$ for the cycle-consistency loss from a small number close to 0 to a maximum number $\lambda_{cyc}^{max}$. $i$ denotes the index of interaction step for Atari and the index of environment step for DMControl. When $i$ is smaller than $i_{end}$, $\lambda_{cyc} = \lambda_{cyc}^{max} \cdot \exp(-5 \cdot (1 - \frac{i}{i_{end}})^2)$ according to a Gaussian ramp-up curve before a warmup end iteration $i_{end}$. Otherwise, $\lambda_{cyc} = \lambda_{cyc}^{max}$. We set $i_{end}$ to 50k. We set $\lambda_{pred} = 1$ and $\lambda_{cyc}^{max} = 1$.

**GPU Setup.** In this work, we run each experiment on one GPU (NVIDIA Tesla V100, P40 or P100).

### A.3   Environment and Code

In this work, we evaluate models on Atari [2] and DMControl [10], which are commonly used benchmarks for discrete and continuous control, respectively. The two benchmarks do not involve personally identifiable information or offensive contents. Our implementation code for Atari is based on SPR [9] assert[3], and that for DMControl is mainly based on CURL [7] assert[4].

### A.4   Error Bar of Main Results

Due to space limitation, we report the error bar (the mean and standard deviation over 10 random seeds) only on DMControl-100k and report the mean scores on Atari-100k. Here, we report the standard deviation over 15 random seeds for both *Baseline* (*i.e.*, SPR run by us) and PlayVirtual on Atari-100k in Table 1. We can see that the standard deviation of our PlayVirtual is comparable with that of *Baseline*.

Table 1: The standard deviation (STD) comparison of *Baseline* and PlayVirtual on Atari-100k. The STD is obtained from 15 runs with random seeds.

| Game | Baseline | PlayVirtual | Game | Baseline | PlayVirtual | Game | Baseline | PlayVirtual |
|---|---|---|---|---|---|---|---|---|
| Alien | 138.8 | 231.7 | Crazy Climber | 6275.9 | 4664.4 | Kung Fu Master | 4095.1 | 6198.7 |
| Amidar | 43.0 | 41.3 | Demon Attack | 207.6 | 332.4 | Ms Pacman | 546.9 | 330.7 |
| Assault | 138.8 | 50.2 | Freeway | 15.3 | 13.9 | Pong | 6.5 | 13.2 |
| Asterix | 229.8 | 170.5 | Frostbite | 1075.0 | 1196.3 | Private Eye | 0.0 | 23.5 |
| Bank Heist | 97.2 | 160.9 | Gopher | 251.9 | 276.6 | Qbert | 1053.2 | 952.6 |
| Battle Zone | 4027.3 | 5261.6 | Hero | 2940.3 | 2130.9 | Road Runner | 3940.8 | 3765.5 |
| Boxing | 13.6 | 19.9 | Jamesbond | 47.3 | 75.3 | Seaquest | 111.9 | 126.9 |
| Breakout | 3.9 | 4.4 | Kangaroo | 3551.8 | 3183.0 | Up N Down | 2848.4 | 10398.1 |
| Chopper Command | 337.0 | 318.7 | Krull | 323.7 | 524.6 | | | |

## B   More Experimental Results and Analysis

### B.1   More Ablation Studies

We present more ablation studies, including effectiveness of PlayVirtual at different environment steps, warmup scheme, weight for cycle consistency loss and where to add the cycle consistency constraint. We use the median HNS of the 26 Atari games and the median score of the 6 DMControl environments to measure the overall performance on Atari and DMControl, respectively. We run each game in Atari with 15 random seeds. To save computational resource, we run each environment in DMControl with 5 random seeds.

**Effectiveness of PlayVirtual at Different Environment Steps.** To further benchmark PlayVirtual's data efficiency, we compare the testing performance in every 5k environment steps at the first 100k

---

[3]Link: `https://github.com/mila-iqia/spr`, licensed under the MIT License.
[4]Link: `https://github.com/MishaLaskin/curl`, licensed under the MIT License.

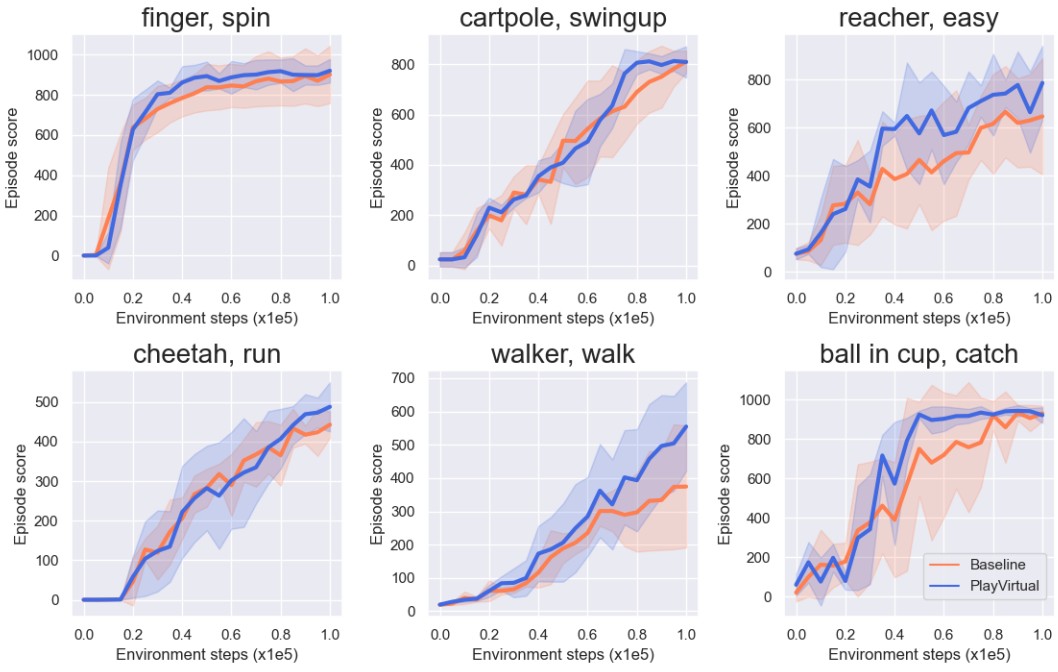

Figure 1: Test performance comparison on DMControl where the lines denote the mean score and the shadow indicates the corresponding standard deviation (obtained by running each environment with 5 random seeds). Our PlayVirtual (marked with blue) outperforms *Baseline* (marked with orange) in most environments by a large margin at different environment steps.

on DMControl. Figure 1 shows the test performance curves of *Baseline* (SPR[†]) and PlayVirtual. We can see that our PlayVirtual performs better than *Baseline* in most environments, where the curves of PlayVirtual outperform *Baseline* by a large margin on "reacher, easy", "walker, wall", and "ball in cup, catch" environments.

**Effectiveness of the Warmup for** $\lambda_{cyc}$**.** Instead of setting $\lambda_{cyc}$ to be a predefined value $\lambda_{cyc}^{max}$, as described in Appendix A.2, we ramp up the weight $\lambda_{cyc}$ in training. We compare the performance of our PlayVirtual without using warmup and with warmup in Table 2, which shows that warmup can benefit the training and results in better performance.

Table 2: Influence of warmup for the weight $\lambda_{cyc}$ w.r.t. the cycle consistency loss.

| Model | Atari-100k | DMControl-100k |
|---|---|---|
| Baseline | 37.1 | 728.0 |
| PlayVirtual(w/o warmup) | 42.5 | 749.5 |
| PlayVirtual | **47.2** | **797.0** |

**Influence of Predefined Weight** $\lambda_{cyc}^{max}$ **w.r.t. the Cycle Consistency Loss.** We set a maximum weight value $\lambda_{cyc}^{max}$ for the cycle consistency loss in the warmup scheme. We study the influence of this hyperparameter in Table 3. We find that $\lambda_{cyc}^{max} = 1$ provides superior performance for both Atari and DMControl.

**Where to Add the Cycle Consistency Constraint?** For the cycle consistency constraint, we can add this constraint at the end step (*i.e.*, $d_{\mathcal{M}}(\mathbf{z}_t', \tilde{\mathbf{z}}_t)$ at $t$) or at every step (*e.g.*, $d_{\mathcal{M}}(\mathbf{z}_t', \tilde{\mathbf{z}}_t) + \sum_{k=1}^{k=K-1} d_{\mathcal{M}}(\mathbf{z}_{t+k}', \hat{\mathbf{z}}_{t+k})$) w.r.t. the backward trajectory (see Figure 1 in our main manuscript for better understanding). Table 4 shows the performance for the two cases. We find their results are similar, where the end-step case is slightly better than the every-step case. A possible explanation is that the estimated states from the DM may be not accurate and the supervision from them in every step

Table 3: Influence of predefined weight $\lambda_{cyc}^{max}$ w.r.t. the cycle consistency loss.

| $\lambda_{cyc}^{max}$ | 0 | 0.1 | 1 | 2 | 10 |
|---|---|---|---|---|---|
| Atari-100k | 37.1 | 40.7 | **47.2** | 45.5 | 41.9 |
| DMC-100k | 723.0 | 777.0 | **797.0** | 740.5 | 763.5 |

(besides the end-step) may bring side-effect. For simplicity, we add the cycle consistency constraint only at the end-step where the state $\tilde{z}_t$ (which is obtained from the observation $s_t$) is reliable.

Table 4: Ablation study on where to add the cycle consistency constraint.

| Model | Atari-100k | DMControl-100k |
|---|---|---|
| Baseline | 37.1 | 728.0 |
| PlayVirtual(every step) | 46.1 | 781.0 |
| PlayVirtual(end step) | **47.2** | **797.0** |

## B.2 Complexity

We compare the complexity of PlayVirtual with *Baseline* in terms of running time and the number of parameters. The inference time of PlayVirtual is exactly the same as *Baseline*, since the network architecture of their encoder and the policy learning head are the same, where the auxiliary task is discarded in test. Averagely, our method increases *Baseline*'s training time by about 6% on Atari and 12% on DMControl, which is acceptable.

PlayVirtual introduces a backward dynamics model on top of *Baseline* in training. PlayVirtual has a very close number of parameters to that of *Baseline* on DMControl. For example, on "cartpole, swingup" (DMControl), PlayVirtual has 25.86M parameters while *Baseline* has 25.81M parameters. On "pong" (Atari), PlayVirtual has 3.91M parameters while *Baseline* has 3.83M parameters.

## C More Discussion

**How Does PlayVirtual Avoid Trivial Solutions in the Latent Space?** Our proposed method does not fall into trivial solutions (such as a constant representation vector) due to the following reasons. (i) We adopt the policy learning (RL) loss to update the encoder to prevent it from falling into this trivial solution. (ii) We also do inference for the dynamics model using real trajectories and supervise the prediction with the representations of the groundtruth states. (iii) We also adopt a target encoder and stop gradient scheme as in SPR [9] and BYOL [3] to avoid the representation collapse.

**Performance of Dynamics Model.** We conduct an evaluation on the dynamics model (DM). Particularly, after 100k environment steps training, we calculate the average prediction mean squared error (MSE) of DM in latent space over 1000 transitions. The evaluation is on a subset of DMControl environments with 5 random seeds. The comparison results of *Baseline* (SPR[†]) and PlayVirtual are shown in Table 5. We can see that our models achieve better prediction performance than *Baseline*. Thanks to our cycle-consistency regularized virtual trajectories generation, we safely augment the trajectories for learning better state representations, which also results in a stronger dynamics model.

Table 5: Evaluation on dynamics models in *Baseline* and our method. The mean squared error (MSE) results of dynamics prediction are reported.

| MSE | Cartpole, swingup | Reacher, easy | Cheetah, run |
|---|---|---|---|
| Baseline | 0.2517 | 0.3920 | 0.0731 |
| PlayVirtual | **0.2357** | **0.3633** | **0.0672** |

**Performance of Learned Representations.** Besides the final performance reported in our main manuscript, we further evaluate the state representations by studying which kind of representations can better promote the policy learning. As shown in Table 6, we consider three schemes. (i) For *None*, models are trained from scratch with only RL loss (*i.e.*, $\mathcal{L}_{rl}$). (ii) For *Baseline Encoder*, models are trained with only RL loss while their encoders are initialized with (100k environment steps) SPR[†]-pretrained encoder parameters, and these encoders are fixed during training. (iii) For *PlayVirtual Encoder*, the setting is similar to (ii) except for initializing the encoders with PlayVirtual-pretrained encoder parameters. We test the 100k-step performance (*i.e.*, scores) on a subset of DMControl environments with 5 random seeds. As shown in Table 6, we can observe that the model whose encoder is initialized by a pretrained *PlayVirtual Encoder* performs better than that of *Baseline Encoder* and non-pretrained non-fixed encoder (*i.e.*, *None*). This observation demonstrates the state representations learned by our method are more helpful to the policy learning.

Table 6: Evaluation on learned representations. The 100k-step scores of models with different pretrained encoders are reported.

| Initialization | Cartpole, swingup | Reacher, easy | Cheetah, run |
|---|---|---|---|
| None | $796 \pm 60$ | $730 \pm 185$ | $388 \pm 89$ |
| Baseline Encoder | $839 \pm 24$ | $517 \pm 141$ | $478 \pm 30$ |
| PlayVirtual Encoder | $\mathbf{847 \pm 31}$ | $\mathbf{828 \pm 67}$ | $\mathbf{512 \pm 31}$ |

**Method of Action Sampling.** In this work, we uniformly sample actions from the action space when generating virtual trajectories. Although the study of action sampling is not the focus of this work, we do evaluate other action sampling methods such as adding zero-mean Gaussian noise $\mathcal{N}(0, \sigma)$ to the original actions in the real trajectories. We conduct the experiment with 5 random seeds. The results in Table 7 show that using uniformly sampled actions (*i.e.*, *Random Action*) achieves higher performance than the above-mentioned Gaussian-noise perturbed actions (*i.e.*, *Perturbed Action ($\sigma$)*). This maybe because random actions can "explore" more states for boosting representation learning. Further, there can be more advanced sampling methods such as surprise-based sampling or policy-guided sampling. We leave the study on them as future work.

Table 7: Study on action sampling methods in generating virtual trajectories. *Perturbed Action ($\sigma$)* denotes adding $\mathcal{N}(0, \sigma)$ Gaussian noise to the original actions, while *Random Action* indicates uniformly sampled actions. We report the median scores across 6 DMControl environments.

| DMControl | Perturbed Action (0.01) | Perturbed Action (0.02) | Perturbed Action (0.05) | Random Action (Ours) |
|---|---|---|---|---|
| Median Score | 732.0 | 747.0 | 764.0 | **797.0** |

**Why Do We Predict Dynamics in the Latent Space?** We predict environment dynamics in the latent space instead of the observation space for two reasons. (i) For high-dimensional control tasks such as image-based RL, we expect to learn compact and informative representations that exclude control-irrelevant information to better serve policy learning. If we stay in the observation space, the representations would include control-irrelevant information to reconstruct some control-irrelevant details, which distracts RL algorithms and slows down the policy learning speed [11]. (ii) Staying in the latent space requires less computational cost as the dimension is lower.

**Application and Limitation.** Our proposed method PlayVirtual, which augments cycle-consistent virtual trajectories, is generic and can be applied to many existing RL frameworks. In this work, we apply it on top of two model-free methods: SPR for discrete control benchmark and on top of a variant of SPR, *i.e.*, SPR[†]for continuous control benchmark. But it is not limited to the two baselines. Our method should be applicable to model-based RL methods to improve data efficiency. We leave the implementation on top of other model-free or model-based baselines as future work. However, our method also bears some limitations such as not excelling in non-deterministic environments where the environment dynamics is difficult to be modeled and the cycle consistency in the forward-backward trajectory may be hard to meet.

## D   Potential Societal Impact

Deep reinforcement learning (RL) has broad applications, including games, robotics, healthcare, dialog systems, *etc*. Learning good feature representations is important for deep RL. However, with limited experience, RL often suffers from data inefficiency for training. In this work, we propose a general method, dubbed PlayVirtual, which augments cycle-consistent virtual trajectories to enhance the data efficiency for RL feature representation learning. We have demonstrated the effectiveness of our PlayVirtual, which achieves the best performance on both discrete control benchmark and continuous control benchmark. We believe our technique will promote the progress of RL applications and inspire more interesting works on improving the data efficiency for RL. Meanwhile, for image-based RL, systems should be developed following responsible AI policies to be fair and safe.

Table 8: Hyperparameters used for Atari.

| Hyperparameter | Value |
| --- | --- |
| Gray-scaling | True |
| Frame stack | 4 |
| Observation downsampling | (84, 84) |
| Augmentation | Random shift & intensity |
| Action repeat | 4 |
| Training steps | 100K |
| Max frames per episode | 108K |
| Reply buffer size | 100K |
| Minimum replay size for sampling | 2000 |
| Mini-batch size | 32 |
| Optimizer | Adam |
| Optimizer: learning rate | 0.0001 |
| Optimizer: $\beta_1$ | 0.9 |
| Optimizer: $\beta_2$ | 0.999 |
| Optimizer: $\epsilon$ | 0.00015 |
| Max gradient norm | 10 |
| Update | Distributional Q |
| Dueling | True |
| Support of Q-distribution | 51 bins |
| Discount factor | 0.99 |
| Reward clipping Frame stack | [-1, 1] |
| Priority exponent | 0.5 |
| Priority correction | $0.4 \rightarrow 1$ |
| Exploration | Noisy nets |
| Noisy nets parameter | 0.5 |
| Evaluation trajectories | 100 |
| Replay period every | 1 step |
| Updates per step | 2 |
| Multi-step return length | 10 |
| Q network: channels | 32, 64, 64 |
| Q network: filter size | $8 \times 8, 4 \times 4, 3 \times 3$ |
| Q network: stride | 4, 2, 1 |
| Q network: hidden units | 256 |
| Target network update period | 1 |
| $\tau$ (EMA coefficient) | 0 |
| **Additional Hyperparameters in PlayVirtual** | |
| K (number of prediction steps) | 9 |
| M (number of virtual trajectories) | $2|\mathcal{A}|$ (two times of action space size) |
| $\lambda_{pred}$ (weight for prediction loss) | 1 |
| $\lambda_{cyc}^{max}$ (a weight related to cycle consistency loss) | 1 |
| Warmup | Gaussian ramp-up ($i_{end}$=50K) |

Table 9: Hyperparameters used for DMControl.

| Hyperparameter | Value |
|---|---|
| Frame stack | 3 |
| Observation rendering | (100, 100) |
| Observation downsampling | (84, 84) |
| Augmentation | Random crop & intensity |
| Replay buffer size | 100000 |
| Initial exploration steps | 1000 |
| Action repeat | 2 finger-spin and walker-walk; |
| | 8 cartpole-swingup; |
| | 4 otherwise |
| Evaluation episodes | 10 |
| Optimizer | Adam |
| $(\beta_1, \beta_2) \rightarrow (\theta_f, \xi_h, \xi_b, \omega)$ | (0.9, 0.999) |
| $(\beta_1, \beta_2) \rightarrow (\alpha)$ (temperature in SAC) | (0.5, 0.999) |
| Learning rate $(\theta_f, \omega)$ | 0.0002 cheetah-run |
| | 0.001 otherwise |
| Learning rate $(\theta_f, \xi_h, \xi_b)$ | 0.0001 cheetah-run |
| | 0.0005 otherwise |
| Learning rate $(\alpha)$ | 0.0001 |
| Policy batch size $(\theta_f, \omega)$ | 512 |
| Auxiliary batch size $(\theta_f, \xi_h, \xi_b)$ | 128 |
| Q-function EMA $\tau$ | 0.01 |
| Critic target update freq | 2 |
| Discount factor | 0.99 |
| Initial temperature | 0.1 |
| Target network update period | 1 |
| Target network EMA $\tau$ | 0.05 |
| **Additional Hyperparameters in PlayVirtual** | |
| K (number of prediction steps) | 6 |
| M (number of virtual trajectories) | 10 |
| $\lambda_{pred}$ (weight for prediction loss ) | 1 |
| $\lambda_{cyc}^{max}$ (a weight related to cycle consistency loss) | 1 |
| Warmup | Gaussian ramp-up ($i_{end}$=50K) |