# OpenReview forum: "PlayVirtual: Augmenting Cycle-Consistent Virtual Trajectories for Reinforcement Learning"
_NeurIPS.cc/2021/Conference — NeurIPS 2021 Poster_

### Official Review · Reviewer_1uRz · 2021-07-13

**Rating:** 6
**Confidence:** 4

**Summary:**

This paper introduces PlayVirtual a new, self-supervised, methodology for use in reinforcement learning settings based on the idea of enforcing cycle-consistency when predicting latent environment representations with (learned) forward/backwards dynamics models. As PlayVirtual is primarily positioned as a means by which to improve data efficiency, the experimental results are targetted to the low-data regime and, following related prior work, report their results on Atari and DeepMind control suites of tasks. For these experiments, the proposed PlayVirtual methodology outperforms competing methods (on average across tasks within the two suites) and thereby obtains SOTA performance.


**Limitations And Societal Impact:**

While the "Potential Societal Impact" section of this paper does feel like an afterthought (being the last section of the appendix) I do agree with the authors that this work has little potential for negative societal impact. I would like to seem more discussion of the limitations of their work.

**Main Review:**

# Post-rebuttal

I appreciate the authors thorough response to several of my concerns and to the questions raised by the other reviewers. I am substantially more confident that the proposed PlayVirtual method's impact on learned representations is meaningful. In my view, the main remaining weakness of this paper is the lack of pedagogical insight as to why and how cycle consistency is improving the representations. Some work has been done in this direction in the rebuttal, I particularly appreciate the experiment with the perturbed actions, but I maintain that the takeaway message from the experiments is still primarily "cycle consistency is good." Nevertheless, the authors responses have made me lean more towards acceptance than rejection and I have updated my rating to reflect this.
____

# Original review
This paper has several clear strengths:

* The proposed PlayVirtual method is intuitive, not overly encumbered by hyperparameters, and is sufficiently generic that it is not difficult to imagine it being used for other reinforcement learning tasks.
* The experiments provide a strong argument that PlayVirtual can result in a large benefit within the low data regime on commonly used benchmarks.
* The idea of using simulated trajectories for representation learning instead of policy improvement is interesting.

Despite these strengths I'm struggling to commit to acceptance.

Overall:

I would be more than happy to change my mind but my general impression is that this work feels quite incremental and struggles to meet the standard for novelty for NeurIPS. In particular: (a) this paper builds heavily on SPR and thus seems to have few if any modeling contributions, (b) the idea of using cycle consistency for representation learning has been exhaustively exploited in other domains, (c) experiments are comprehensive but not pedagogical, I learn little more than "cycle-consistency is good", and (d) there seems to be little consideration of more realistic/challenging tasks. It thus seems that the main novelty of this work is the third strength that I noted above: the use of simulated trajectories for representation learning. If this is the case I would expected to have some of the following questions answered:

* What is the effect of action sampling on the learned representations? Is it better to sample trajectories from the learned policy, randomly, or from some "teaching" policy?
* How are the encodings learned through cycle consistency different from those learned simply using SPR? What are the types of behaviors that your learned agents exhibit that SPR models don't?
* How would you extend PlayVirtual to more complicated environments, especially PO-MDPs where we'd expect deterministic forward/backward prediction to break down?
* How does the performance/capacity of the dynamics models impact the learned representations?

I would expect to increase my rating if some portion of the above questions were answered well or if I could be convinced that the novelty was more substantial than I initially understood.

Minor line-by-line comments:

Line 22
* I believe you mean "data inefficient" rather than "data efficient"?


Line 23-28
* It's a bit odd to motivate you work referencing complex settings where collecting data is hard and then apply your model in a space where we can easily collect huge amounts of data.


Line 37
* "using 100k interaction data" reads a bit weird. I'd rephrase as "allowed 100k interactions with the environment" or similar.

Line 50-56
* A priori, all of the losses you introduce have an easy optimal solution: predict z_t to be a non-zero constant vector. It's worth reiterating why this isn't an issue (as covered in the SPR paper).


Line 159
* "has few opportunity" -> "has few opportunities"


Lines 214-220 (Definition 1)
* I'm not sure what this "mathematical definition" is bringing the paper, indeed it seems to raise more questions than it answers:
  * What is the expectation over (formally)?
  * Why bother with an expectation here at all? It seems like you're effectively just saying that d(z_t', z_t) should be identically 0 over the support.


Lines 224-231
* I find this discussion to be inadequate. This is especially true for the comment regarding many-to-one transitions which I struggle to believe.


Line 244
* In DMControl, are environment and interaction steps equivalent?


**Time Spent Reviewing:**

5.5

---

> ### Author Response · Authors · 2021-08-10
> **Response to Reviewer 1uRz (Part 1)**
>
> Thank you for your helpful comments! We especially appreciate that you think that our method is "sufficiently generic" and "the idea of using simulated trajectories for representation learning instead of policy improvement is interesting".
>
> We understand your concerns. We would like to respond to your overall comments before answering the questions. We hope that the following can address your main concerns.
> - Novelty. The key novelty of our work is the idea of generating virtual trajectories for promoting feature representation learning, instead of using cycle consistency. The use of cycle consistency is to help us ensure the reasonableness of these trajectories.
> - Broad applicability. Our method can be applied to most control tasks as long as the environment dynamics can be modeled. From a methodological point of view, it is easy to apply our method to many models, not just SPR, as pointed out by Reviewer ddpW, "*Proposed idea is quite general (independent of environment, domain, continuous or discrete. image or state) which might be useful in other fields like robotics*". We claimed in Appendix. C that we will leave the implementation on top of other baselines as future work. Although Atari and DMControl are widely used benchmarks in RL, we hope to consider more realistic/challenging tasks beyond Atari and DMControl in the future work.
>
> We now answer your questions individually.
> - **Q1: "What is the effect of action sampling on the learned representations? Is it better to sample trajectories from the learned policy, randomly, or from some "teaching" policy?"**
>
>   **A1**: The action sampling affects the final performance but is not the focus of our work. We study using real trajectories (“Baseline+BDM” scheme), as shown in Table 3 in the main text, where the gain is smaller than random actions (i.e., uniformly sampled actions). This is because “Baseline+BDM” is of limited diversity of trajectories. In addition, we also tried policy-guided action sampling (i.e., using "Perturbed Action ($\sigma$)" where we add Gaussian noise $N(0, \sigma$) on the original actions to generate the executed actions). As shown in the table below, measured by the median score results (averaged over 5 random trials) on DMControl, there is no significant improvements for using policy-guided action sampling (Perturbed Action) when compared to the random sampling.
>
>     |DMControl| Perturbed Action (0.01) | Perturbed Action (0.02) | Perturbed Action (0.05)| Random Action
>     | :-----| :----: | :----: | :----: | :----: |
>     | Median Score | 732.0 | 747.0 | 764.0 | 797.0 |
>
>   The idea of sampling actions from some "teaching" policies is interesting where something-guided/driven action sampling may be more effective. We will consider this in the future work. We will discuss the points in detail in our revision.
>
> $~$
> - **Q2: "How are the encodings learned through cycle consistency different from those learned simply using SPR? What are the types of behaviors that your learned agents exhibit that SPR models don't?"**
>
>   **A2**: The encodings/representations learned by ours help the policy learn more efficiently than the representations of SPR. This is because we generate rich and reasonable virtual trajectories to regularize and improve both the encoder and DM, as illustrated in the ablation in Line 337-355, Section 4.3. We made some extensive evaluations on the learned representations and DM on a subset of DMControl environments with 5 random seeds. We hope the results below will address some of your concerns.
>
>   - **State representation evaluation**. Besides the final performance reported in our main paper, we further evaluate the state representations by studying which kind of representations can better promote the policy learning. Specifically, we initialize the encoder with different pretrained (100k-step) encoder parameters (note that we fix the initialized encoder during the training) and train other network parameters from scratch, then we test the 100k-step performance (i.e., scores) on a subset of DMControl environments with 5 random seeds. As shown in the table below, we can observe that the model whose encoder is initialized by a pretrained PlayVirtual Encoder performs better than that of SPR Encoder and non-pretraining (i.e., None). This observation demonstrates the state representations learned by our proposed method are more helpful to the policy learning.
>     |Initialization| Cartpole, swingup | Reacher, easy | Cheetah, run
>     | :-----| :----: | :----: | :----: |
>     | None         | 796 ± 60 | 730 ± 185 | 388 ± 89 |
>     | SPR Encoder         | 839 ± 24 | 517 ± 141 | 478 ± 30 |
>     | PlayVirtual Encoder | **847 ± 31** | **828 ± 67** | **512 ± 31** |
>   - **Dynamics model evaluation**. We conducted an evaluation on DM: after 100k environment steps training, we calculate the average prediction mean squared error (MSE) of DM over 1000 transitions. The evaluation is on a subset of DMControl environments with 5 random seeds. The comparison results of SPR and PlayVirtual are as follows. We can see that our model is of higher prediction accuracy than SPR in the MSE metric. This observation indicates that, through cycle-consistency regularized virtual trajectories generation, we safely augment the trajectories for learning better state representations yielding a stronger dynamics model.
>
>     |MSE| Cartpole, swingup | Reacher, easy | Cheetah, run
>     | :-----| :----: | :----: | :----: |
>     | SPR         | 0.2517 | 0.3920 | 0.0731 |
>     | PlayVirtual | **0.2357** | **0.3633** | **0.0672** |
>
> $~$
> - **Q3: "How would you extend PlayVirtual to more complicated environments, especially PO-MDPs where we'd expect deterministic forward/backward prediction to break down?"**
>
>   **A3**: As with most methods that model environmental dynamics [1 - 7], it will increase the risk of use in this case. We will clarify this in our revision.
>
>   Reference:
>
>   [1] Shelhamer, E., Mahmoudieh, P., Argus, M., and Darrell, T. Loss is its own reward: Self-supervision for reinforcement learning. ArXiv, abs/1612.07307, 2017.
>
>   [2] Hafner, D., Lillicrap, T., Fischer, I., Villegas, R., Ha, D., Lee, H., and Davidson, J. Learning latent dynamics for planning from pixels. In International Conference on Machine Learning, pp. 2555–2565. PMLR, 2019.
>
>   [3] Hafner, D., Lillicrap, T., Ba, J., and Norouzi, M. Dream to control: Learning behaviors by latent  imagination. In International Conference on Learning Representations, 2020.
>
>   [4] Łukasz Kaiser, Babaeizadeh, M., Miłos, P., Osin´ski, B., Campbell, R. H., Czechowski, K., Erhan, D., Finn, C., Kozakowski, P., Levine, S., Mohiuddin, A., Sepassi, R., Tucker, G., and Michalewski, H. Model based reinforcement learning for atari. In International Conference on Learning Representations, 2020.
>
>   [5] Guo, Z. D., Pires, B. A., Piot, B., Grill, J.-B., Altché, F., Munos, R., and Azar, M. G. Bootstrap latent-predictive representations for multitask reinforcement learning. In International Conference on Machine Learning, pp. 3875–3886. PMLR, 2020.
>
>   [6] Lee, K., Seo, Y., Lee, S., Lee, H., & Shin, J. Context-aware dynamics model for generalization in model-based reinforcement learning. In International Conference on Machine Learning (pp. 5757-5766). PMLR, 2020.
>
>   [7] Schrittwieser J, Antonoglou I, Hubert T, et al. Mastering atari, go, chess and shogi by planning with a learned model[J]. Nature, 2020, 588(7839): 604-609.
>
> $~$
> - **Q4: "How does the performance/capacity of the dynamics models impact the learned representations?"**
>
>   **A4**: A good dynamics model can predict the future more accurately, so as to  regularize the representations to include more accurate future information. Additionally, we empirically verify this. Our dynamics model (DM) is more accurate than that of SPR as shown in the above DM comparison, and when we do not let the cycle loss update DM (i.e., "PlayVirtual-ND" model in Table 5 in the text), the gain of PlayVirtual drops from 10.1% to 6.9%, as shown in the ablation in Line 337-355. This shows that good dynamics model helps the representations.

---

> > ### Author Response · Authors · 2021-08-10
> > **Response to Reviewer 1uRz (Part 2)**
> >
> > Response to "Minor line-by-line comments":
> > - **Q5: "Line 22: I believe you mean "data inefficient" rather than "data efficient"?"**
> >
> >   **A5**: It should be "data inefficient". We will revise it.
> >
> > $~$
> > - **Q6 : "Line 23-28: It's a bit odd to motivate you work referencing complex settings where collecting data is hard and then apply your model in a space where we can easily collect huge amounts of data."**
> >
> >   **A6**: Thank you for the careful comments! We limit the number of interactions to simulate the difficulty of collecting data in a real environment as much as possible, just like most previous work. We will clarify it in the revision.
> >
> > $~$
> > - **Q7: "Line 37: "using 100k interaction data" reads a bit weird. I'd rephrase as "allowed 100k interactions with the environment" or similar."**
> >
> >   **A7**: Thanks for pointing this out. We will rephrase it per your recommendations.
> >
> > $~$
> > - **Q8: "Line 50-56: "A priori, all of the losses you introduce have an easy optimal solution: predict z_t to be a non-zero constant vector. It's worth reiterating why this isn't an issue (as covered in the SPR paper)."**
> >
> >   **A8**: Thank you for the suggestions! We will add the corresponding analysis in the revision. Similar to SPR, we prevent the dynamics model to map z_t to a non-zero constant vector from the following aspects: (1) We adopt the policy learning (RL) loss to update the encoder to prevent it from the representation collapse; (2) We also do inference for the dynamics model using real trajectories and supervise the prediction with ground-truth states; (3) We also adopt a target encoder and stop gradient scheme as in SPR and BYOL [1] to avoid the representation collapse.
> >
> >   Reference:
> >
> >   [1] Grill, J.-B., Strub, F., Altché, F., Tallec, C., Richemond, P., Buchatskaya, E., Doersch, C.,  Avila Pires, B., Guo, Z., Gheshlaghi Azar, M., Piot, B., kavukcuoglu, k., Munos, R., and Valko, M. Bootstrap your own latent - a new approach to self-supervised learning. In Advances in Neural Information Processing Systems, 2020.
> >
> > $~$
> > - **Q9: "Line 159: "has few opportunity" -> "has few opportunities""**
> >
> >   **A9**: Thanks for pointing it out. We will correct it in the revision.
> >
> > $~$
> > - **Q10: "Lines 214-220 (Definition 1): "What is the expectation over (formally)?  Why bother with an expectation here at all? It seems like you're effectively just saying that d(z_t', z_t) should be identically 0 over the support."**
> >
> >   **A10**: Sorry for the confusion. The expectation is over trajectory cycle (i.e., forward-backward trajectory). We originally use the expectation to formulate the cycle consistency to generally consider both deterministic dynamics models (as we use in this paper) and probabilistic dynamics models. To be more clear, we will follow your valuable suggestion to say that d(z_t', z_t) should be identically 0 in our revied version.
> >
> > $~$
> > - **Q11: "Lines 224-231: I find this discussion to be inadequate. This is especially true for the comment regarding many-to-one transitions which I struggle to believe."**
> >
> >   **A11**: Thanks for pointing this out. We would like to expand the discussion and limitation, and make some qualitative/quantitative analysis about the non-deterministic environments in the revision.
> >
> > $~$
> > - **Q12: "Line 244: In DMControl, are environment and interaction steps equivalent?"**
> >
> >   **A12**: No. Environment steps are equal to the product of interaction steps and action repeat (i.e., "environment steps = interaction steps * action repeat").
> >
> > $~$
> > - **Q13 (Limitations And Societal Impact): "While the "Potential Societal Impact" section of this paper does feel like an afterthought (being the last section of the appendix) I do agree with the authors that this work has little potential for negative societal impact. I would like to seem more discussion of the limitations of their work."**
> >
> >   **A13**: Thanks for your suggestions. We will supplement the limitations from at least two aspects: applicable models and risk of use in non-deterministic environments.

---

> ### Author Response · Authors · 2021-08-22
> **Further reply**
>
> Thank you for your further review!
>
> The cycle consistency improves the state representations in two aspects: (1) The cycle consistency plays a role as the supervision signals for the generated virtual trajectories and ensures their reasonableness. With the help of these safe/reasonable virtual trajectories, we can enrich the "experience" of the feature encoder. (2) The cycle consistency is also a regularization when training the dynamics models.
>
> We perform a specific ablation to study which of the above is more important in Line 313-319 in Section 4.3. The result indicates that the former dominates the achieved improvements. We will analyze this in more detail to provide more and clearer pedagogical insights in the revision.

---

### Official Review · Reviewer_aagy · 2021-07-16

**Rating:** 6
**Confidence:** 4

**Summary:**

RL algorithms are generally data-inefficient and would benefit from representation learning methods as augmented loss for better data efficiency. Building on top of SPR, this paper proposes using timewise cycle-consistency in the latent space to augment the training data. Specifically, the forward and backward latent dynamics model is trained using supervised learning from the ground truth. In addition, a cycle-consistency loss is imposed on states with randomly sampled actions for N steps.

The paper experimented on the Atari 100k and DMControl 100k benchmarks and showed SOTA performance in terms of Median HNS for Atari and Median score for DMControl.


**Limitations And Societal Impact:**


1. What is the weakness of your method? How can a user choose when to use it and when not to use it? What about potential future directions?
2. How does environmental randomness come into play for cycle consistency? Does your method work equally well for deterministic and stochastic environments (either inherent to the env like Kung Fu Master or via sticky actions)?

**Main Review:**

While cycle consistency is not new, it was used previously in RL mainly as a way for Sim2real ([Rao et al. 2020](https://arxiv.org/abs/2006.09001), [Zhang et al. 2021](https://arxiv.org/abs/2012.09811)). The authors proposed a new method to incorporate cycle consistency as a regularizer in the latent representations of the states for RL. The results on Atari and DMControl benchmarks are promising and are worth further investigation. The proposed method is simple to implement and does not add much additional cost in terms of training time.


While the strength of this simple yet effective incremental improvement to SPR is clear, I believe to increase the impact of this paper, the readers would benefit from a more detailed discussion of one or more of the following points:
1. Good feature representations and data efficiency are related, but not entirely the same. The method proposed claims to help with both, but the experimental result only supports better data efficiency. A good representation can help with: a) OOD generalization and extrapolation b) human interpretability c) Faster learning speed. Is it possible to have experiments to support the claim that cycle consistency leads to better feature representation in RL (e.g. using the criteria in [Anand et al. 2019](https://arxiv.org/abs/1906.08226))? Alternatively you can claim that it only helps with data efficiency.
2. In 4.2, following the format in SPR, can you also report the mean HNS as well as # of games with super human performance for your best agent here? Did your agent beat the baselines in all metrics or just the median?
3. Have you tried non-random actions for the cycle consistency loss?
4. Can you elaborate why you chose to stay in the latent space vs in the observation or action space? It’s quite an important design choice and I can see the pros and cons for both.


Other suggestions to improve the readability of this paper
1. In the paper, it is unclear whether the PlayVirtual predicts the raw state that the agents get from the env, or the latent := Encoder(state). From the implementation, it looks like the latter, but please update the rest of the document to distinguish the raw state/observation from the latent state. (e.g. l7-9, l44-47, etc.)
2. From figure 1, it looks like the encoder takes 4 frames for $s_t$. Is that just the batch, or is that actually the frames $s_{t-3} \sim s_t$?
3. For DMControl, [Laskin et al. 2020](https://arxiv.org/abs/2004.14990) reports a performance on-par with PlayVirtual. So the claim in l14-15 is inaccurate.


## Updates after rebuttal

The authors’ rebuttals made me believe more deeply that there is something interesting with the empirical results. It's good that they also added more about the limitations and future work. More follow up analysis and a better understanding of why it worked would make the paper much stronger.

I’m happy to change my recommendation to accept in order to encourage future work and more analysis on this empirically strong finding, especially around whether it works in more environments and whether it is a general technique that works beyond SPR.

I have changed my ratings accordingly.

**Time Spent Reviewing:**

6

---

> ### Author Response · Authors · 2021-08-10
> **Response to Reviewer aagy**
>
> Thank you for your valuable feedback! We especially appreciate that you think that our method is "simple yet effective".
>
> We address your questions and concerns below.
>
> - **Q1: "Good feature representations and data efficiency are related, but not entirely the same. The method proposed claims to help with both, but the experimental result only supports better data efficiency. A good representation can help with: a) OOD generalization and extrapolation b) human interpretability c) Faster learning speed. Is it possible to have experiments to support the claim that cycle consistency leads to better feature representation in RL (e.g. using the criteria in Anand et al., 2019)? Alternatively you can claim that it only helps with data efficiency."**
>
>   **A1**: Thanks for your valuable suggestion. In this paper, we generate virtual trajectories to regularize feature representation learning for improving the data efficiency. In such a context, we tend to use the data efficiency (i.e., faster learning speed) to reflect the quality of feature representations. We will further clarify this as suggested in our revision.
>
> $~$
> - **Q2: "In 4.2, following the format in SPR, can you also report the mean HNS as well as # of games with super human performance for your best agent here? Did your agent beat the baselines in all metrics or just the median?"**
>
>   **A2**: Median HNS is one of the most commonly used metrics on Atari games [1-4]. We will add more results with more metrics as in the SPR paper (1. Median HNS, 2. Mean HNS, 3. Median DQN@50M-Normalized Score (DQN@50M-NS), 4. Mean DQN@50M-NS, 5. # Games Superhuman). SPR (Reported): results reported by SPR paper with 10 random seeds; SPR (Reproduced): our reproduced results using SPR official source code with 15 random seeds. PlayVirtual (with 15 random seeds) outperforms SPR (Reported/Reproduced) in most metrics, as shown in the table below.
>
>     |Atari | Medain HNS (%) | Mean HNS (%)| Median DQN@50M-NS (%) | Mean DQN@50M-NS (%) | # Games Superhuman
>     | :-----| :----: | :----: | :----: | :----: | :----: |
>     | SPR (Reported)   | 41.5 | 70.4 | 36.1 | 51.0 | 7 |
>     | SPR (Reproduced) | 37.1 | 61.4 | 38.2 | 56.5 | 4 |
>     | PlayVirtual      | 47.2 | 63.7 | 37.4 | 65.0 | 4 |
>
>   Reference:
>
>   [1] Laskin M, Srinivas A, Abbeel P. Curl: Contrastive unsupervised representations for reinforcement learning. In International Conference on Machine Learning. PMLR, 2020: 5639-5650.
>
>   [2] Yarats, D., Kostrikov, I., and Fergus, R. Image augmentation is all you need: Regularizing deep reinforcement learning from pixels. International Conference on Learning Representations, 2021.
>
>   [3] Liu G, Zhang C, Zhao L, et al. Return-Based Contrastive Representation Learning for Reinforcement Learning. In International Conference on Learning Representations. 2020.
>
>   [4] Schwarzer, M., Anand, A., Goel, R., Hjelm, R. D., Courville, A., and Bachman, P. Data-efficient reinforcement learning with self-predictive representations. In International Conference on Learning Representations, 2021.
>
> $~$
> - **Q3: "Have you tried non-random actions for the cycle consistency loss?"**
>
>   **A3**: Yes, we have tried non-random actions. We study actions from real trajectories (i.e., original actions) for the cycle consistency loss, i.e., “Baseline+BDM” scheme in Table 3 in the text. We find that this scheme brings marginal gains, which indicates the gain of PlayVirtual mainly lies in the augmentation of “virtual” actions. We further consider using both original and “virtual” actions for the cycle loss, but we haven’t observed an obvious improvement. For example, on Atari, the median HNS of using both actions is 47.3% while PlayVirtual is 47.2%. In future work, we would like to consider more types of actions.
>
> $~$
> - **Q4: "Can you elaborate why you chose to stay in the latent space vs in the observation or action space? It’s quite an important design choice and I can see the pros and cons for both."**
>
>   **A4**: We chose to stay in the latent space for two reasons: (1) For high-dimensional control tasks such as image-based RL, we hope to learn compact and informative representations that exclude control-irrelevant information, so as to better serve policy learning. If we stay in the observation space, the representations need to include control-irrelevant information to reconstruct some control-irrelevant details, which "distracts" RL algorithms and slows down the policy learning speed [1]. Additionally, prior work has proven that latent dynamics models can effectively abstract observations to predict forward in compact state [2 - 5]. (2) Staying in the latent space requires less computational cost as the dimension is smaller.
>
>   Reference:
>   [1] Zhang, A., McAllister, R. T., Calandra, R., Gal, Y., and Levine, S. Learning invariant representations for reinforcement learning without reconstruction. In International Conference on Learning Representations, 2021.
>
>   [2] M. Watter, J. Springenberg, J. Boedecker, and M. Riedmiller. Embed to control: A locally linear latent dynamics model for control from raw images. In Advances in Neural Information Processing Systems, pages 2746–2754, 2015.
>
>   [3] J. Oh, S. Singh, and H. Lee. Value prediction network. In Advances in Neural Information Processing Systems, pages 6118–6128, 2017.
>
>   [4] K. Gregor, D. J. Rezende, F. Besse, Y. Wu, H. Merzic, and A. v. d. Oord. Shaping belief states with generative environment models for rl. arXiv preprint arXiv:1906.09237, 2019.
>
>   [5] Hafner, D., Lillicrap, T., Ba, J., and Norouzi, M. Dream to control: Learning behaviors by latent imagination. In International Conference on Learning Representations, 2020.
> ---
> Response to “Other suggestions to improve the readability of this paper”:
> - **Q5: "In the paper, it is unclear whether the PlayVirtual predicts the raw state that the agents get from the env, or the latent := Encoder(state). From the implementation, it looks like the latter, but please update the rest of the document to distinguish the raw state/observation from the latent state. (e.g. l7-9, l44-47, etc.)"**
>
>   **A5**: Sorry for this confusion. The prediction is performed on the latent states. We will make it clearer and update the rest of the document in our revision.
>
> $~$
> - **Q6: "From figure 1, it looks like the encoder takes 4 frames for $s_t$. Is that just the batch, or is that actually the frames $s_{t-3} \sim s_t$?"**
>
>   **A6**: A state such as $s_t$ is a stack of several consecutive frames. The number of stacked frame ("Frame stack") is 4 for Atari and 3 for DMcontrol, as shown in the hyperparameters tables in Appendix.
>
> $~$
> - **Q7: "For DMControl, Laskin et al. 2020 reports a performance on-par with PlayVirtual. So the claim in l14-15 is inaccurate."**
>
>   **A7**: Thank you for pointing it out. We will make modifications to this claim in our revision.
>
> $~$
> - **Q8 (Limitations And Societal Impact): "What is the weakness of your method? How can a user choose when to use it and when not to use it? What about potential future directions? How does environmental randomness come into play for cycle consistency? Does your method work equally well for deterministic and stochastic environments (either inherent to the env like Kung Fu Master or via sticky actions)?"**
>
>   **A8**: PlayVirtual has no guarantee for performance improvement when encountering many-to-one transitions as we discuss in Line 224-231. In this work, we show that PlayVirtual can improve the baseline (i.e. SPR) on widely used benchmarks, Atari and DMControl where the environmental dynamics is deterministic. We leave the exploration on non-deterministic environments to our future work. Further, we plan to apply the idea of PlayVirtual to other baselines and fields. We will also leave the study of action sampling strategies to our future work.

---

> > ### Comment · Reviewer_aagy · 2021-08-15
> > **Further comments**
> >
> > Dear authors,
> >
> > Thank you for addressing my questions and suggestions.
> >
> > **Q4: prediction over the latent space vs the observation space.**
> >
> > Your response to Q4 makes sense. It’d be nice to add the rationale to the appendix or to the main body.
> >
> > I am struggling to recommend acceptance mainly because I am concerned by the lack of insights into the learned latent. This concern motivated me to ask Q1 and Q4. I thought predicting in the observation space directly is one way to make it more interpretable. But as reviewer ddpW pointed out, there are other ways to interpret the learned latent like doing a NN search in the dataset.
> >
> > I am worried that with a lack of understanding of the learned latent and/or the conditions under which cycle consistency will work, it would make future work hard to do. Other researchers working in a new domain will need to try PlayVirtual, and if it doesn’t perform well, they don’t have many options apart from 1. Tuning the number of steps to predict into the future 2. Give up PlayVirtual and try something else.
> >
> > Looking forward to hearing more thoughts on this.

---

> > > ### Author Response · Authors · 2021-08-18
> > > **Further reply**
> > >
> > > Dear reviewer,
> > >
> > > Thank you for your reply!
> > >
> > > We initially evaluated the quality of the latent states predicted by our DM and BDM through measuring the distance between our predictions and the groundtruth latent states using the mean squared error (MSE). The corresponding results are provided in our response to the Q1 of Reviewer ddpW, which indicates that our method has better quality of the predicted/learned latent states.
> > >
> > > Here, we also conduct a nearest neighbor (NN) search evaluation for the latent states as suggested. We perform a k-NN search by taking the predicted results as the queries to retrieve the groundtruth states and report the accuracies (mean & std) of different values of k (i.e., k=1, k=5, k=10) over 1000 test transitions with 5 random seeds on Cheetah-Run, DMControl below. As shown in the table below, the k-NN search evaluation provides consistent comparison results with the MSE metric, further demonstrating the quality of our predicted/learned latent states. Thus, we can believe that our model does not fall into trivial or meaningless solutions.
> > >
> > >   |Retrieval Accuracy (%) | 1-NN | 5-NN| 10-NN |
> > >   | :-----| :----: | :----: | :----: |
> > >   | SPR | 57.4 ± 24.3 | 78.9 ± 16.7 | 87.6 ± 11.5 |
> > >   | PlayVirtual | 79.8 ± 9.6 | 93.8 ± 5.0 | 97.4 ± 2.5 |
> > >
> > > Worthy of mention is that the state representations are learned for the policy learning rather than the reconstruction in the pixel domain [1]. Therefore, states with the same control-relevant information (such as the foreground) but different control-irrelevant information (such as the background) may be mapped to similar representations in the latent space. This many-to-one mapping hinders the retrieval.
> > >
> > >
> > > Reference:
> > >
> > > [1] Zhang, A., McAllister, R. T., Calandra, R., Gal, Y., and Levine, S. Learning invariant representations for reinforcement learning without reconstruction. In International Conference on Learning Representations, 2021.

---

> ### Author Response · Authors · 2021-08-27
> **Thank you**
>
> Thank you for your great efforts and valuable suggestions! We appreciate your kind support, and we will try our best to improve our text based on your comments.

---

### Official Review · Reviewer_ddpW · 2021-07-17

**Rating:** 7
**Confidence:** 5

**Summary:**

The paper proposes a method to learn better representations for reinforcement learning tasks. They propose augmenting the feature encoder with two dynamics models: 1) forward dynamics model that predicts future latent state given an action and the current latent state 2) backward dynamics model that takes the current latent state and previous action to predict the previous latent state. They use these 2 models to apply 2 losses: 1) future latent state prediction loss 2) cycle-consistency constraint that ensures that if you iteratively predict the future latent state K times into the future using the forward model and K times back into the past using the backward model, the difference between the initial latent state and predicted latent state should be minimal.

They show empicirally that these additional losses improve performance of an RL algorithm on multiple benchmarks (Atari and DMControl).  They have an ablation study that shows how important each of the auxiliary losses are on both benchmarks.

**Limitations And Societal Impact:**

Yes the authors have adequately addressed societal impact and limitations of their work

**Main Review:**

Strengths
1) Extensive experiments on multiple RL benchmarks using strong baselines shows that cycle-consistency constraints result in better features which leads to better performance.
2) Nice ablation study showing the importance of each loss.
3) Proposed idea is quite general (independent of environment, domain, continuous or discrete. image or state) which might be useful in other fields like robotics.
4) Well organized paper with enough information to reproduce the work.

Weakness
1. Analysis of Virtual Trajectories
Since these trajectories are in the latent space, it is not clear what the latent space is actually encoding. The model can always project a latent state into the "future" by trivially mapping it to a constant. This makes backward mapping "in time" also easy. This is probably not happening as adding the losses is improving performance. But having some statistics to measure that the predictions done by forward and backward model are meaningful will be nice (how much does the latent state z change as it forward and backward projected). Some visualization might also be useful. Can the authors take an initial frame and the forward model, predict K time steps into the future and use that predicted embedding to get the nearest-neighbor frame from the dataset. Comparison with initial frame and this retrieved frame should give some insight into what is being modeled by the dynamics model. Can the authors use different actions to show the forward and backward models are sensitive to the input action or do they ignore the action?

2. How are actions sampled while applying the loss?
The authors state "In this way, by augmenting actions (generating/sampling virtual actions), we can obtain abundant virtual cycle-consistent trajectories for training". Do they randomly sample many actions while training the forward backward model? How do the authors obtain the latent for unseen actions as it seems you can only apply loss to seen trajectories as the true z^tilde_t+k comes from the observed trajectory?

3. Virtual trajectories not used in RL loss
The proposed virtual trajectories are not used directly for training the RL algorithm i.e. the auxiliary losses only use the augmented trajectories. This is concerning because the authors state that "we enforce a trajectory to meet the cycle consistency constraint, which can significantly enhance the data efficiency". As the authors show this constraint is applied after the agent has interacted many times with the environment. Without these initial interaction steps it is difficult to learn a good forward/backward model. While the authors generate trajectories in the latent space these are not used by the RL loss which makes it difficult to motivate that the proposed approach will be significantly sample efficient?
Can the authors use these virtual trajectories in the RL loss? My concern is that while the performance improvement might due to the auxiliary losses but it might not mean that the virtual trajectories will always be more sample-efficient as they somehow "enrich the experience". "PlayVirtual" title might suggest agent is virtually interacting with environment but that does not seem to be the case. One explanation might be that the additional constraints help in learning better representations by preventing overfitting to one objective.

4. Do the auxiliary losses have to be applied iteratively?
This is not a big weakness but more of a curiosity from my side. From Table 4 the authors show the best performance is achieved for predicting K=9 steps into the future. Is it possible to then only apply  a one forward prediction that is K=9 steps in the future. They can concat the actions of next 9 steps and provide that as input to the forward model. If similar performance is achieved then it might save some training time by preventing the iterative process.

5. Training curves showing how the auxiliary losses change as training proceeds.

6. What is RDM in line 345? Is it supposed to BDM?

Justification of Rating

This paper involved a lot of experiments across many different environments. The authors did a good job of explaining the method and validating various design decisions empirically. The paper's quality will improve if they add the missing details (asked in weakness section) and justify the use of the term VirtualPlay as it seems the policy is not really using these virtual trajectories directly. Even though their proposed approach improves performance, it is important to establish the exact cause why it does so? Is it better representations or are the "virtual trajectories" actually adding more simulated experience to the agent? Currently there is only evidence for the former in the paper.

Missing citations
1) Connection with the loss used in [25]
Dwibedi, D., Aytar, Y., Tompson, J., Sermanet, P., & Zisserman, A. (2019). Temporal cycle-consistency learning. In Proceedings of the IEEE/CVF Conference on Computer Vision and Pattern Recognition (pp. 1801-1810).

**Time Spent Reviewing:**

4

---

> ### Author Response · Authors · 2021-08-10
> **Response to Reviewer ddpW**
>
> Thank you for the valuable comments! We especially appreciate that you think that our idea is "general" and "might be useful in other fields".
>
> We address your questions and concerns below.
>
> - **Q1: Will the model project a latent state into the "future" by trivially mapping it to a constant? What is the performance of DM and BDM? Are they sensitive to the input actions?**
>
>   **A1**: No, our proposed method won't fall into this trivial solution due to the following reasons: (1) We adopt the policy learning (RL) loss to update the encoder to prevent it from falling into this trivial solution; (2) We also do inference for the dynamics model using real trajectories and supervise the prediction with groundtruth states; (3) We also adopt a target encoder and stop gradient scheme as in SPR and BYOL [1] to avoid the representation collapse. Besides, we experimentally observe the predicted latent states, and find that there is no tendency for the latent states to be mapped into a constant.
>
>   We conducted the following additional experiments to address your remaining questions. The evaluation is on a subset of DMControl environments with 5 random seeds.
>   - **DM and BDM performance evaluation**. We conducted the evaluation as follows: after 100k environment steps training, we calculate the average prediction mean squared error (MSE) of DM over 1000 transitions (where the actions are produced by the policy, denoted as “original action”). We report the results of our DM and BDM as well as SPR’s DM (as a reference) in the table below.  The BDM’s forward-then-backward prediction MSE (cooperated with DM) is larger than the DM’s forward prediction MSE, which is understandable due to the accumulation of error.
>   - **The sensitivity test of DM and BDM on input action**. We tested this by disturbing actions produced by the policy with white noise (±0.3) below (denoted as “disturbed action”). We found that the MSEs of DM and BDM tend to increase when we disturb the input actions, which verifies the meaningfulness of the latent states to a certain extent.
>     |MSE (original / disturbed action)| Cartpole, swingup | Reacher, easy | Cheetah, run
>     | :-----| :----: | :----: | :----: |
>     | SPR's DM| 0.2517 / 0.2548 | 0.3920 / 0.4000 | 0.0731 / 0.0727 |
>     | Our DM | 0.2357 / 0.2361 | 0.3633 / 0.3678| 0.0672 / 0.0691
>     | Our BDM | 0.3622 / 0.3854 | 0.5929 / 0.6152 | 0.1117 / 0.1129
>
>   Reference:
>
>   [1] Grill, J.-B., Strub, F., Altché, F., Tallec, C., Richemond, P., Buchatskaya, E., Doersch, C.,  Avila Pires, B., Guo, Z., Gheshlaghi Azar, M., Piot, B., kavukcuoglu, k., Munos, R., and Valko, M. Bootstrap your own latent - a new approach to self-supervised learning. In Advances in Neural Information Processing Systems, 2020.
>
> $~$
> - **Q2: "How are actions sampled while applying the loss? ... How do the authors obtain the latent for unseen actions as it seems you can only apply loss to seen trajectories as the true z^tilde_t+k comes from the observed trajectory?"**
>
>   **A2**: We uniformly sample actions from the action space. For the unseen actions, we train the backward dynamics model to form a cycle and apply the cycle consistency loss to provide supervisions. In this way, we don't need to obtain the groundtruth of the intermediate states as we only perform the cycle-consistency supervision on the end latent state $\mathbf{z}'_t$ in the trajectory cycle.
>
> $~$
> - **Q3: "Virtual trajectories not used in RL loss. The proposed virtual trajectories are not used directly for training the RL algorithm i.e. the auxiliary losses only use the augmented trajectories. This is concerning because the authors state that "we enforce a trajectory to meet the cycle consistency constraint, which can significantly enhance the data efficiency". As the authors show this constraint is applied after the agent has interacted many times with the environment. Without these initial interaction steps it is difficult to learn a good forward/backward model. While the authors generate trajectories in the latent space these are not used by the RL loss which makes it difficult to motivate that the proposed approach will be significantly sample efficient? Can the authors use these virtual trajectories in the RL loss? My concern is that while the performance improvement might due to the auxiliary losses but it might not mean that the virtual trajectories will always be more sample-efficient as they somehow "enrich the experience."**
>
>   **A3**: These generated virtual trajectories enhance the data efficiency by regularizing the state representation learning, where the cycle-consistency constraint is to ensure the reasonableness of these trajectories. The cycle loss on virtual trajectories is applied together with the prediction loss on the real trajectories and is warmed up by adjusting its corresponding weights (see Line 51-57 in Appendix). We don't directly use these virtual trajectories for policy learning due to the absence of rewards. We will further clarify these in the revision.
>
> $~$
> - **Q4: ""PlayVirtual" title might suggest agent is virtually interacting with environment but that does not seem to be the case. One explanation might be that the additional constraints help in learning better representations by preventing overfitting to one objective."**
>
>   **A4**: We are sorry for this confusion. Our dynamics model is trained to simulate the state transition process of the real environment, which can be viewed as a surrogate of the environment in terms of dynamics. Thus, the process of generating virtual trajectories by feeding sampled actions into the surrogate environment (i.e., the dynamics model) can be considered as a kind of virtual interaction. We will further clarify this to avoid the ambiguity in our revision.
>
> $~$
> - **Q5: "Do the auxiliary losses have to be applied iteratively? This is not a big weakness but more of a curiosity from my side. From Table 4 the authors show the best performance is achieved for predicting K=9 steps into the future. Is it possible to then only apply a one forward prediction that is K=9 steps in the future. They can concat the actions of next 9 steps and provide that as input to the forward model. If similar performance is achieved then it might save some training time by preventing the iterative process."**
>
>   **A5**: No, it doesn’t have to be. But we would like to argue that applying auxiliary losses could provide more dense and accurate supervision signals, which makes the optimization eaiser.
>
>   We have considered the higher-order dynamics model when designing PlayVirtual and empirically found that the performance of the high-order dynamic model is not as good as the first-order dynamic model.
>
> $~$
> - **Q6: "Training curves showing how the auxiliary losses change as training proceeds."**
>
>   **A6**: Thank you for the suggestions! We will consider it in the revision.
>
> $~$
> - **Q7: "What is RDM in line 345? Is it supposed to BDM?"**
>
>   **A7**: Thanks for pointing this out. The "RDM" in line 345 should be "BDM" and we will correct it in the revision.
>
> $~$
> - **Q8: Missing citation.**
>
>   **A8**: Thanks for pointing it out. We will cite this paper in the revision.
>
>   Dwibedi, D., Aytar, Y., Tompson, J., Sermanet, P., & Zisserman, A. (2019). Temporal cycle-consistency learning. In Proceedings of the IEEE/CVF Conference on Computer Vision and Pattern Recognition (pp. 1801-1810).

---

> > ### Comment · Reviewer_ddpW · 2021-08-27
> > **Response to Response**
> >
> > Thanks for the answers to my questions. I am inclined to accept the paper given that the authors satisfactorily answered my questions.
> >
> > I am a bit confused about the experimental setup of the nearest neighbor experiment mentioned in the reply to reviewer aagy. What is the query embedding, what is the dataset over which the search is done and what is the accuracy of k-NN when no cycle-consistency loss is applied?

---

> > > ### Author Response · Authors · 2021-08-27
> > > **Further response**
> > >
> > > Thank you!
> > >
> > > We use the predicted latent state as the query embedding while leaving the rest of the states in the dataset as the galleries. The dataset is comprised of trajectories sampled from the environment through the learned model. When there is no cycle-consistency loss, our model degenerates into SPR whose results are provided in the reply to Reviewer aagy.

---

> > > > ### Comment · Reviewer_ddpW · 2021-08-29
> > > > **Further Response**
> > > >
> > > > Thanks for the reply.
> > > >
> > > > I wanted a clarification about the "rest of the states in the dataset as the galleries". Are these states obtained by running the final policy at the end of training with N different start states or is this a single trajectory? My concern is that there can be similar states in multiple trajectories but the k-NN classifier is considering each state from different trajectories as same. What is the final size of the gallery? Additionally this is an interesting result in its own right that the learned predicted states can be used to recover the true state when so many similar states exist in the gallery. The results of ~80% accuracy top-1 NN  implies the representations learned are quite good. Hence I wanted to know what the size of the gallery might be. Any visualization of what the k-NN retrieves v/s ground truth would also help a bit more in understanding this result.

---

> > > > > ### Author Response · Authors · 2021-08-30
> > > > > **Reply to Further Response**
> > > > >
> > > > > Thank you for your reply!
> > > > >
> > > > > The states in the dataset are obtained by running the final policy at the end of training (with N=4 different start states). This is to say that we collect 4 real trajectories, where each trajectory includes 250 states (leaving the size of gallery to be 1000 in total), and we retrieve the predicted states from this gallery. We make some visualizations on the retrieval results and find that our method can successfully retrieve the groundtruth state in most cases even if there are many similar states. We will add the visualization results in the revision.

---

> > > > > > ### Comment · Reviewer_ddpW · 2021-08-30
> > > > > > **Reply to Reply to Further Response**
> > > > > >
> > > > > > Thanks so much for the clarification. These extra details will help the readers understand the representations learned due to the additional cycle-consistency loss.

---

> > > > > > > ### Author Response · Authors · 2021-08-31
> > > > > > > **Thank you**
> > > > > > >
> > > > > > > Thank you! We will try our best to improve our text based on your comments.

---

### Official Review · Reviewer_Lrfb · 2021-07-20

**Rating:** 6
**Confidence:** 3

**Summary:**

This paper introduces PlayVirtual, an approach for learning better representations by training an encoder to learn cycle-consistent forward and backward dynamics models. This differs from model-based approaches in that the virtual trajectories are used to train a representation rather than as experience for training a policy. The approach is evaluated within Atari and DM-control tasks.

**Limitations And Societal Impact:**

There is not discussion of the limitations of the approach. However, there is some discussion about the feasibility of the backwards dynamics model. In particular, it could be the case that in a given state $s_{t+1}$, the same action $a$ could have been taken in different states $s_t$. I.e., there might not be a unique state that $b(s_{t+1}, a_t)$ would map to.

**Main Review:**

Overall, I found the paper to be clearly written and well-motivated. While forward and backwards dynamics models have been used for generating experiences for training policies, I thought the idea of using these models to aid in creating unseen trajectories for training a representation was interesting and novel. Several works have used forward dynamics models for training representations, but these methods typically require ground-truth state and action pairs. So being able to sample from the action space to generate new experiences is quite useful.

However, I had some doubts in terms of the empirical results. While PlayVirtual achieves higher HNS in the Atari experiments, it performs worse than the baseline that it builds upon, SPR, in 12/26 of the environments. Furthermore, there are no results for SPR in the DM-Control tasks showing individual performance for each task, which makes it difficult to determine if the improved performance was from the PlayVirtual model or SPR. The paper would be greatly improved by including these results in Table 2.

Comments:
- I found the description of the model in Figure 1 and throughout the text to be somewhat confusing. In particular, the original actions are not augmented but rather a new trajectory is formed by sampling from the action space. I think it would make the approach more clear to describe in this way.
- Was the accuracy of the learned dynamics models measured?
- I’m curious how different action sampling methods could improve performance. For example, sampling based on “surprise” as done in model-based methods rather than uniformly sampling.


**Time Spent Reviewing:**

4.5

---

> ### Author Response · Authors · 2021-08-10
> **Response to Reviewer Lrfb**
>
> Thank you for the valuable comments! We especially appreciate your positive comments on our motivation, novelty as well as method design.
>
> We address your questions and concerns below.
>
> - **Q1: Empirical results on Atari.**
>
>   **A1**: The HNS is one of the most widely used evaluation metric on Atari games following [1 - 7]. HNS reflects the level of the agents relative to humans on Atari games. Higher median HNS over the 26 Atari games indicates higher overall statistical performance.
>   $~$
>
>   Reference:
>
>   [1] Mnih V, Kavukcuoglu K, Silver D, et al. Human-level control through deep reinforcement learning[J]. Nature, 2015, 518(7540): 529-533.
>
>   [2] Schrittwieser J, Antonoglou I, Hubert T, et al. Mastering atari, go, chess and shogi by planning with a learned model[J]. Nature, 2020, 588(7839): 604-609.
>
>   [3] Badia A P, Piot B, Kapturowski S, et al. Agent57: Outperforming the atari human benchmark. In International Conference on Machine Learning. PMLR, 2020: 507-517.
>
>   [4] Laskin M, Srinivas A, Abbeel P. Curl: Contrastive unsupervised representations for reinforcement learning. In International Conference on Machine Learning. PMLR, 2020: 5639-5650.
>
>   [5] Yarats, D., Kostrikov, I., and Fergus, R. Image augmentation is all you need: Regularizing deep reinforcement learning from pixels. International Conference on Learning Representations, 2021.
>
>   [6] Liu G, Zhang C, Zhao L, et al. Return-Based Contrastive Representation Learning for Reinforcement Learning. In International Conference on Learning Representations. 2020.
>
>   [7] Schwarzer, M., Anand, A., Goel, R., Hjelm, R. D., Courville, A., and Bachman, P. Data-efficient reinforcement learning with self-predictive representations. In International Conference on Learning Representations, 2021.
>
> $~$
> - **Q2: "Furthermore, there are no results for SPR in the DM-Control tasks showing individual performance for each task, which makes it difficult to determine if the improved performance was from the PlayVirtual model or SPR. The paper would be greatly improved by including these results in Table 2."**
>
>   **A2**: Thanks for this suggestion. The individual performance (100k steps, 5 random seeds) of SPR for each task on DMcontrol is shown in Figure 1 in Appendix. We will move it to our main paper. Here we give the numerical results of SPR and PlayVirtual on DMControl with 10 random seeds. We will include the numerical results of SPR in Table 2 of our main paper.
>
>     |DMControl-100k| Finger, spin | Cartpole, swingup |Reacher, easy | Cheetah, run | Walker, walk | Ball in cup, catch | Median Score|
>     | :-----| :----: | :----: | :----: | :----: | :----: |:----: | :----: |
>     | SPR         | 868 ± 143 | 799 ± 42 | 638 ± 269 | 467 ± 36 | 398 ± 165 | 861 ± 233 | 719.0 |
>     | PlayVirtual | **915 ± 49** | **816 ± 36** | **785 ± 142** | **474 ± 50** | **460 ± 173** | **926 ± 31** | **800.5** |
>
>     |DMControl-500k| Finger, spin | Cartpole, swingup |Reacher, easy | Cheetah, run | Walker, walk | Ball in cup, catch | Median Score|
>     | :-----| :----: | :----: | :----: | :----: | :----: |:----: | :----: |
>     | SPR         | 924 ± 132 | **870 ± 12** | 925 ± 79 | 716 ± 47 | 916 ± 75 | 963 ± 8 | 920.0 |
>     | PlayVirtual | **963 ± 40** | 865 ± 11 | **942 ± 66** | **719 ± 51** | **928 ± 30** | **967 ± 5** | **935.0** |
>
> $~$
> - **Q3: "I found the description of the model in Figure 1 and throughout the text to be somewhat confusing. In particular, the original actions are not augmented but rather a new trajectory is formed by sampling from the action space. I think it would make the approach more clear to describe in this way."**
>
>   **A3**: We are sorry for the confusion caused by our description. The action augmentation lies in the level of the entire action set, referring to increasing the diversity of actions instead of modifying the original actions. We will clarify this in our revision based on your suggestions.
>
> $~$
> - **Q4: "Was the accuracy of the learned dynamics models measured?"**
>
>   **A4**: Yes. We conducted an evaluation on the learned dynamics models: after 100k environment steps training, we calculate the average prediction mean squared error (MSE) of DM over 1000 transitions. The evaluation is on a subset of DMControl environments with 5 random seeds. As illustrated in the table below, the dynamics model in our proposed method is more accurate than that in SPR, thanks to the augmentation of trajectories.
>
>     |MSE| Cartpole, swingup | Reacher, easy | Cheetah, run
>     | :-----| :----: | :----: | :----: |
>     | SPR         | 0.2517 | 0.3920 | 0.0731 |
>     | PlayVirtual | **0.2357** | **0.3633** | **0.0672** |
>
> $~$
> - **Q5: "I’m curious how different action sampling methods could improve performance. For example, sampling based on “surprise” as done in model-based methods rather than uniformly sampling."**
>
>   **A5**: Thanks for your insightful comments. The action sampling affects the final performance but is not the focus of our work. We will add the related discussion in our revision. Note that our core idea can work with the plainest sampling method, i.e., uniform random sampling. We experimentally tried different action sampling methods such as policy-guided action sampling (e.g., "Perturbed Action ($\sigma$)": adding Gaussian noise $N(0, \sigma$) to the original actions) when we designed PlayVirtual. We haven't observed obvious advantages over random(uniform) sampling, as shown in the table below. The "surprise"-based action sampling is very interesting, and intuitively, it can improve performance through some appropriate designs. We will point out the potential of further improving our proposed method by designing more effective sampling methods and leave it to our future work.
>
>     |DMControl| Perturbed Action (0.01) | Perturbed Action (0.02) | Perturbed Action (0.05)| Random Action
>     | :-----| :----: | :----: | :----: | :----: |
>     | Median Score | 732.0 | 747.0 | 764.0 | 797.0 |
>
> $~$
> - **Q6 (Limitations And Societal Impact): "There is not discussion of the limitations of the approach. However, there is some discussion about the feasibility of the backwards dynamics model. In particular, it could be the case that in a given state st+1, the same action could have been taken in different states $s_t$. I.e., there might not be a unique state that $b(s_{t+1}, a_t)$ would map to."**
>
>   **A6**: Thanks for this suggestion, and we will discuss the limitation of our method more in the revision. Our proposed method does not excel in policy learning in non-deterministic environments due to that the dynamics is difficult to be modeled.

---

### Author Response · Authors · 2021-08-22
**Please let us know if our response has addressed your concerns**

We thank all the reviewers for their great efforts in reviewing our paper! We are looking forward to knowing whether our response has well addressed your concerns.

We tried our best to address the main concerns raised in the reviews including (1) providing the evaluation results of our dynamics models as well as further explanations that our method does not fall into trivial solutions, (2) addressing the concerns on the impact of action sampling and (3) further clarifying the reasonableness of our learned latent states and the reasons for choosing to performing dynamics predictions in the latent space.

Please feel free to leave your comments if you have any further questions or suggestions.

---

### Decision · Program_Chairs · 2021-09-28

**Decision:**

Accept (Poster)

**Comment:**

After the author rebuttal and discussion, the reviewers have unanimously agreed to accept the paper. All three reviewers expressed gratitude for the improved clarification of the method and follow-up experiments and the overall empirical value of the presented experiments, but still expressed some reservations along the lines of "The paper would clearly be much stronger if it provided more insight as to why and in what way this cycle consistency loss improves the learned representations". For this reason, I recommend accepting this paper as a poster.


**Consistency Experiment:**

NeurIPS has a long history of experimentation. In 2014, NeurIPS ran an experiment in which 10% of submissions were reviewed by two independent committees to quantify the randomness in the review process. This year, we repeated a variant of this experiment to see how the quality of the review process has changed over time.  This paper was part of the experiment and was therefore assigned to two committees (consisting of reviewers, an Area Chair, and a Senior Area Chair) that reached independent decisions.  If both committees made the same recommendation, this recommendation was followed. If a single committee recommended acceptance, the paper was accepted (with the exception of a few cases in which the other committee identified what we considered a fatal flaw, e.g., an error in a key result).

Both committees reached the same decision: **Accept (Poster)**

The other committee assigned to the paper recommended **Accept (Poster)**.  You can find the other set of reviews, along with any follow up discussion with the authors here:
https://openreview.net/forum?id=GSHFVNejxs7